# A tunable dual-input system for on-demand dynamic gene expression regulation

Elisa Pedone[1,2]*, Lorena Postiglione[1,2,11], Francesco Aulicino[3,4,11], Dan L. Rocca[1,2,3]*, Sandra Montes-Olivas[1], Mahmoud Khazim[1,2], Diego di Bernardo [5], Maria Pia Cosma[6,7,8,9,10] & Lucia Marucci[1,2,3]*

Cellular systems have evolved numerous mechanisms to adapt to environmental stimuli, underpinned by dynamic patterns of gene expression. In addition to gene transcription regulation, modulation of protein levels, dynamics and localization are essential checkpoints governing cell functions. The introduction of inducible promoters has allowed gene expression control using orthogonal molecules, facilitating its rapid and reversible manipulation to study gene function. However, differing protein stabilities hinder the generation of protein temporal profiles seen in vivo. Here, we improve the Tet-On system integrating conditional destabilising elements at the post-translational level and permitting simultaneous control of gene expression and protein stability. We show, in mammalian cells, that adding protein stability control allows faster response times, fully tunable and enhanced dynamic range, and improved in silico feedback control of gene expression. Finally, we highlight the effectiveness of our dual-input system to modulate levels of signalling pathway components in mouse Embryonic Stem Cells.

[1] Department of Engineering Mathematics, University of Bristol, Bristol BS8 1UB, UK. [2] School of Cellular and Molecular Medicine, University of Bristol, Bristol BS8 1TD, UK. [3] BrisSynBio, Bristol BS8 1TQ, UK. [4] Department of Biochemistry, Bristol BS8 1TD, UK. [5] Telethon Institute of Genetics and Medicine Via Campi Flegrei 34, 80078 Pozzuoli, Italy. [6] Centre for Genomic Regulation (CRG), Dr Aiguader 88, 08002 Barcelona, Spain. [7] Universitati Pompeu Fabra (UPF), Barcelona, Spain. [8] ICREA, Pg. Luis Companys, 08010 Barcelona, Spain. [9] Guangzhou Regenerative Medicine and Health Guangdong Laboratory (GRMH-GDL), 510005 Guangzhou, China. [10] Key Laboratory of Regenerative Biology and Guangdong Provincial Key Laboratory of Stem Cells and Regenerative Medicine, Guangzhou Institutes of Biomedicine and Health, Chinese Academy of Science, 510530 Guangzhou, China. [11]These authors contributed equally: Lorena Postiglione, Francesco Aulicino. *email: elisa.pedone@bristol.ac.uk; dan.rocca@bristol.ac.uk; lucia.marucci@bristol.ac.uk

A number of perturbation approaches have been developed to study gene function in biological systems, and for gene therapy applications. It has become increasingly clear that gene expression patterns in vivo are fast and highly dynamic processes, encoding important time-dependent information that underlies many aspects of cellular behaviour[1]. Thus, precise temporal control, as well as reversible manipulation of exogenous gene expression, are fundamental for interrogating cellular functions[2].

Temporal control of gene expression can be achieved by transcriptional regulation via inducible promoters[3–6]. The most widely used system for transcriptional regulation is the Tet system, which consists of the tetracycline transcriptional activator (tTA) and the DNA operator sequence (tetO)[7–11]. The tTA is a fusion protein of the herpes simplex virus VP16 activation domain and of the *Escherichia coli* Tet repressor protein (TetR)[8]. The presence of tetracycline or its derivative doxycycline prevents the interaction of the tTA with the tetO, blocking gene expression (Tet-Off system). The reverse-tTA (rtTA) is a tTA variant allowing gene expression activation in presence of an inducer; the resulting Tet-On system is generally preferred when rapid and dynamic gene induction is required[3,12]. A major limitation of inducible promoters is the significant time delay in switching proteins OFF and ON when using Tet-On and Tet-Off systems, respectively[13], diminishing the possibility of using these approaches to generate dynamic patterns of gene expression that faithfully recapitulate those observed natively[1]. Slow kinetic responses are also common to other techniques targeting precursor DNA or mRNA molecules (e.g. RNA interference[14]), likely due to significantly different rates of innate protein degradation[14].

Recently, an approach relying on conditional protein destabilization to modulate turnover by the cellular degradation machinery has been harnessed to probe biological functions. Engineered mutants of FKBP12 that are rapidly and constitutively degraded in mammalian cells can directly confer destabilisation to the protein they are fused with. The addition of synthetic ligands that bind the Destabilising Domain (DD) of FKBP12 prevents degradation and so can be used to alter levels of the fused-protein of interest[15].

Whilst significantly enhancing the switch-off kinetics as compared to Tet-On, conditional protein regulation systems do not allow independent control of both transcription and translation, which would be highly desirable when studying the correlation between protein and cognate mRNA levels under different spatial and temporal scales[16].

To overcome these limitations, here we present a fully tuneable dual-input system, which allows orthogonal and conditional control at both transcriptional and post-translational levels of a gene of interest. Specifically, we combine a third generation Tetracycline-Inducible System (Tet-On 3 G)[11,17] for inducible and reversible transcriptional regulation with a module incorporating an improved DD from ecDHFR[18] for targeted protein degradation. We demonstrate that our system permits far greater control of both protein dynamics and expression dynamic range across different culture platforms, including microfluidics used for in silico feedback control, and mammalian cell lines. Moreover, we develop an ordinary differential equation model capturing the enhanced dynamic response to inducers. The efficacy of conditional dual-input regulation is exemplified by the ability to incorporate different genes of interest, such as fluorescent proteins and Wnt pathway components, in complex cellular chassis (e.g. mouse embryonic stem cells), paving the way for dynamically controlling mammalian cell behaviour and fate.

## Results

**Dual-input orthogonal regulation of gene expression.** We engineered a mouse Embryonic Stem Cell (mESC) line to stably express the reverse tetracycline transcriptional activator construct (rtTA) and a stable mCherry (henceforth EF1a-rtTA_-TRE3G-mCherry; Fig. 1a), or conditionally destabilised DDmCherry (henceforth EF1a-rtTA_TRE3G-DDmCherry; Fig. 1c) under the control of a TRE3G promoter, which transcribes the gene of interest only in presence of the tetracycline analogue doxycycline[11] (Doxy; Fig. 1a, c). Post-translational control is achieved by applying the small molecule trimethoprim (TMP), which stabilises the destabilising domain (DD)-fused protein in a dose-dependent manner[18]. These two constructs allow for comparison of the standard Tet-On with the dual-input Tet-On/DD system we developed.

We exposed EF1a-rtTA_TRE3G-mCherry and EF1a-rtTA_TRE3G-DDmCherry mESCs to saturating concentrations of Doxy (1000 ng/mL) and TMP (100 nM) for 24 h and checked for mCherry mRNA (Fig. 1b, d, left panels) and protein (Fig. 1b, d, right panels) levels. In absence of Doxy, both transcription and protein expression were undetectable, demonstrating no promoter leakiness (Fig. 1b, d; −Doxy and −Doxy/+TMP samples, respectively), whereas mCherry was robustly activated at both mRNA and protein levels following Doxy and TMP administration (Fig. 1b, d; +Doxy and +Doxy/±TMP, respectively). Notably, Doxy and TMP selectively control transcription and protein stability, respectively: when TMP is combined with Doxy, mCherry mRNA is unchanged (Fig. 1d; left panel), whereas a clear increase in protein stability is observed (Fig. 1d; right panel).

To quantitatively estimate the effect of inducers on protein stability, we measured protein half-life in both EF1a-rtTA_TRE3G-mCherry and EF1a-rtTA_TRE3G-DDmCherry mESCs. Cells were cultured in presence of Doxy and Doxy/TMP for 14 h, and then plated in presence of the protein synthesis inhibitor cycloheximide with varying combinations of Doxy and TMP for 8 h; mCherry was measured by western-blot over the time-course (Fig. 1e). In EF1a-rtTA_TRE3G-mCherry mESCs, untagged mCherry showed no response to inducers, and the half-life was similar across conditions (Fig. 1f; Supplementary Fig. 1a). Of note, mCherry mRNA decreased following Doxy withdrawal (Supplementary Fig. 1c), confirming no effect of Doxy or TMP on protein stability in this system. In contrast, the half-life of the conditionally destabilised mCherry increased approximately three fold in presence of TMP, independently of Doxy addition (Fig. 1g; Supplementary Fig. 1b; +Doxy/+TMP and −Doxy/+TMP samples). mRNA levels decreased only when Doxy was removed (Supplementary Fig. 1d), further demonstrating the specific and independent effect of the two inducers.

Consistently, we found that MG132 blockade of the proteasome[19] enhanced DDmCherry stability, suggesting that TMP acts by preventing proteasomal degradation of DD-fused proteins only[18] (Supplementary Fig. 1e). Indeed, both MG132 and TMP treatment increased DDmCherry abundance, whereas levels of the native β-catenin protein, whose degradation is proteasome mediated[20], increased only when proteasomal activity was inhibited (Supplementary Fig. 1e). Analysing the polyubiquitination status of DDmCherry, following inducer removal and chasing with or without TMP for 12 h, indicated that TMP likely limits addition of ubiquitin chains or promotes de-ubiquitination of DD-tagged proteins, ultimately preventing proteasomal degradation (Supplementary Fig. 1f).

Altogether, these data show that our system allows robust control of both gene transcription and protein stability, with undetectable leakiness and specific response to inducers.

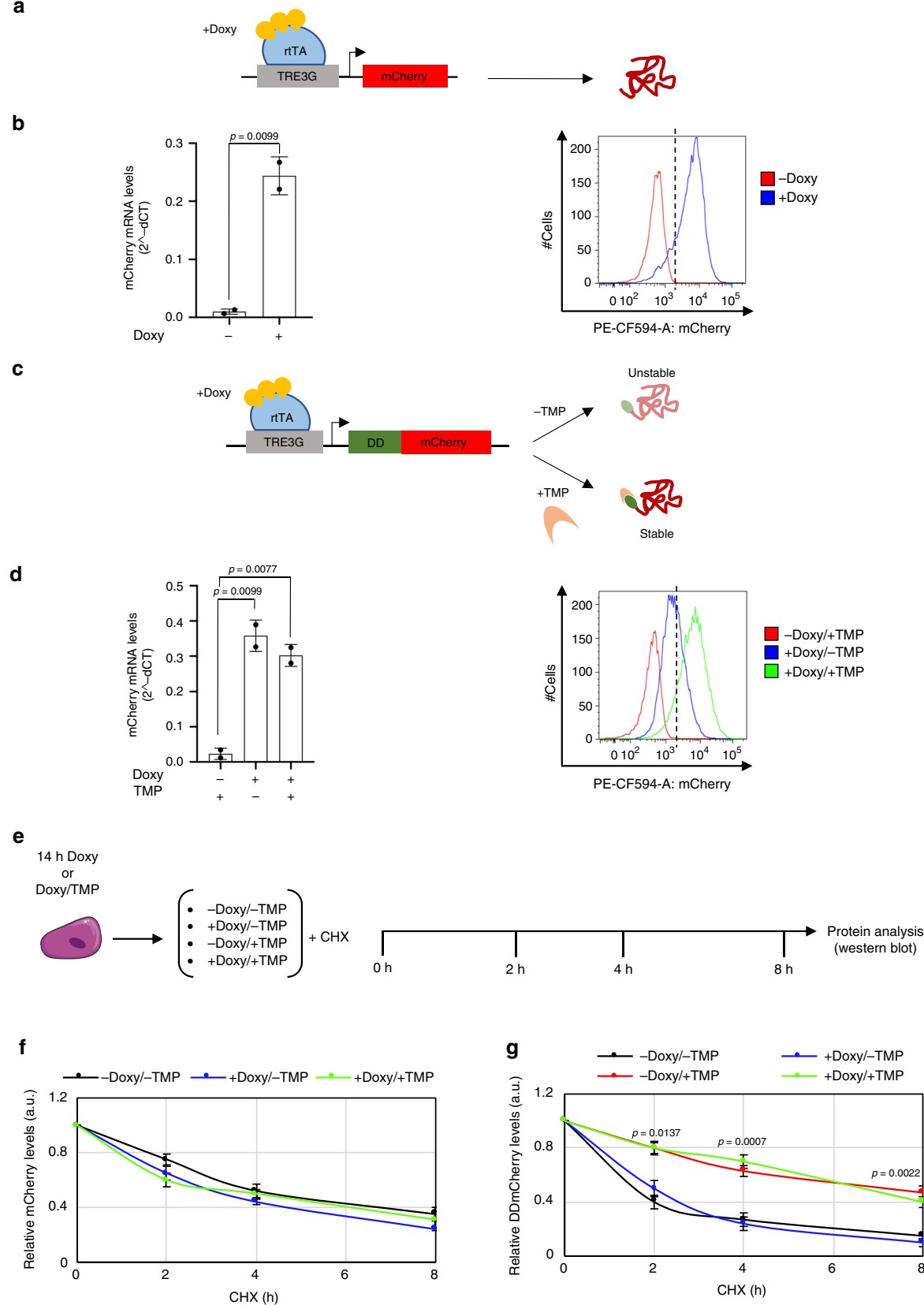

**Dual-input system tunability and dynamic response**. To gauge the tunability and sensitivity of our inducible system, as well as its suitability for dynamic modulation of a gene of interest, we performed both switch-on steady-state titration experiments and switch-off time-courses upon drug administration and removal, in EF1a-rtTA_TRE3G-mCherry and EF1a-rtTA_TRE3G-DDmCherry mESCs.

EF1a-rtTA_TRE3G-mCherry mESCs incubated with six different concentrations of Doxy for 24 h and analysed by flow-cytometry showed a robust response to increasing amount of

**Fig. 1** Dual-input regulation of exogenous protein expression. **a–d** Dual-input regulation system consisting of the reverse transactivator (rtTA) and a stable (**a**) or conditionally destabilised (**c**) mCherry fluorescent protein. mRNA (**b**, **d**, left panels) and protein levels (**b**, **d**, right panels) measured in EF1a-rtTA_TRE3G-mCherry (**b**) and EF1a-rtTA_TRE3G-DDmCherry (**d**) mESCs treated for 24 h with Doxy (1000 ng/mL) or Doxy/TMP (1000 ng/mL and 100 nM, respectively). **e** Experimental scheme of protein half-life measurement. Following 14 h of treatment with Doxy (1000 ng/mL) or Doxy/TMP (1000 ng/mL and 100 nM, respectively), EF1a-rtTA_TRE3G-mCherry and EF1a-rtTA_TRE3G-DDmCherry mESCs were cultured in presence of the protein synthesis inhibitor cycloheximide (CHX, 25 µg/mL) and combination of Doxy (1000 ng/mL), TMP (100 nM) and Doxy/TMP. Protein half-life was measured by western blot after the indicated times of treatment. **f**, **g** Western blot densitometric quantification of EF1a-rtTA_TRE3G-mCherry (**f**) and EF1a-rtTA_TRE3G-DDmCherry (**g**) mESCs. Plotted mCherry and DDmCherry values are normalised against the housekeeping gene GAPDH. Data are means ± SEM (n = 2, **b** and **d**; n = 3, **f** and **g**). p values from two-tailed unpaired t test (**b**, **d**) and nonparametric one-way ANOVA (**f**, **g**) are shown. Source data are provided as a Source Data file

inducer, with saturation reached at Doxy 100 ng/mL (Fig. 2a, dots; Supplementary Fig. 2a); EF1a-rtTA_TRE3G-DDmCherry mESCs showed similar steady-state response when kept with maximal concentration of TMP (10 µM) and varying Doxy in the same range (Fig. 2b, dots; Supplementary Fig. 2b). Also, TMP showed a robust dose-dependent effect when varied while keeping Doxy at maximal concentration (Fig. 2c, dots; Supplementary Fig. 2b).

To complement these experiments, we developed a mathematical model to capture the behaviour of the dual-input system: we relied on ordinary differential equations (ODEs), commonly used to model interactions among genes and other relevant processes, such as mRNA/protein degradation, basal promoter activity and mRNA translation[21,22]. The mathematical models for the EF1a-rtTA_TRE3G-mCherry and EF1a-rtTA_TRE3G-DDmCherry systems are based on sets of three ODEs, describing transactivator and fluorescent gene concentrations as the result of production and degradation terms, and inducer action (Supplementary Note 1). The TET system was modelled, as previously proposed, using Hill kinetics to represent the effect of the inducer on trascription[23]. Given the observed saturating response to TMP (Fig. 2c, dots; Supplementary Fig. 2b and[18]), a Hill function, dependent on TMP, was incorporated in the DDmCherry protein degradation term (Supplementary Note 1). To estimate parameters (Supplementary Table 1 and Note 1), the two ODE models were fitted to the aforementioned steady-state switch-on experimental data and showed good agreement (Fig. 2a–c, dashed lines, Root Mean Squared Error -RMSE- in Fig. 2a–c legends).

Next, we used the fitted models to predict the switch-off dynamic response of both the Tet-On and the Tet-On/DD systems upon inducer withdrawal; model simulations indicated much faster switch-off of the dual-input system (Fig. 2d, e, dashed lines). These predictions were validated experimentally: both EF1a-rtTA_TRE3G-mCherry and EF1a-rtTA_TRE3G-DDmCherry mESCs were able to reach full steady-state induction upon 14 h of incubation with Doxy and Doxy/TMP, respectively, with the conditionally destabilised mCherry showing 80% protein reduction 8 h after inducer removal, as compared to 40% reduction only in the Tet-On system (Fig. 2d, e, dots; Supplementary Fig. 2c, d). When EF1a-rtTA_TRE3G-DDmCherry mESCs were induced with Doxy only, the switch-off dynamics upon Doxy removal were faster than in the EF1a-rtTA_TRE3G-mCherry cells (Supplementary Fig. 2c, d); this is expected, as the DDmCherry protein, in absence of TMP, has lower half-life than mCherry (Fig. 1f, g). However, this also caused cells to present lower mCherry expression upon activation, thus reducing the dynamic range upon induction (Supplementary Fig. 2d).

To assess the response to drugs at single cell level, we analysed the switch-on and -off transitions of 13 individual clones FACS-sorted from both EF1a-rtTA_TRE3G-mCherry and EF1a-rtTA_TRE3G-DDmCherry populations (Supplementary Fig. 2e, f, respectively). Both the switch-on and -off responses were more

homogenous across clones in EF1a-rtTA_TRE3G-DDmCherry cells, as indicated by the mCherry-expressing cell coefficient of variation (Supplementary Fig. 2g); this suggests that conditional control of protein stability might reduce cell-to-cell variability.

These results indicate that the dual-input system we developed allows fully tunable protein induction with both drugs, and faster switch-off dynamics as compared to a standard Tet-On system. Furthermore, the model satisfactory replicated experimental data, indicating that Hill kinetics suit modelling Destabilising Domain responses, for which a mathematical formalism has been missing.

**Modular control and dynamic range.** Given the improved dynamic response of the conditionally destabilised mCherry in EF1a-rtTA_TRE3G-DDmCherry mESCs, we reasoned about further exploiting the performance of the inducible system by fusing the rtTA with a Destabilising Domain. Therefore, we generated two additional mESC lines constitutively expressing a conditionally destabilised version of the rtTA in combination with either a stable or a conditionally destabilised mCherry (Supplementary Fig. 2h, i; EF1a-DDrtTA_TRE3G-mCherry and EF1a-DDrtTA_TRE3G-DDmCherry, respectively).

Both cell lines responded to Doxy/TMP treatment at steady-state, and Doxy and TMP titration experiments indicated good response of mCherry to both drugs (Supplementary Fig. 2j–m, p and q). As expected, when DD is present on both the transactivator (rtTA) and the fluorescent reporter, the residual protein stability when the mRNA is transcribed is reduced (compare +Doxy/−TMP samples in Supplementary Fig. 2b, q).

The fitting of two new ODE systems, describing EF1a-DDrtTA_TRE3G-mCherry and EF1a-DDrtTA_TRE3G-DDmCherry mESC dynamics consistently with the aforementioned modelling assumptions (Supplementary Note 1 and Table 2), showed good agreement with experimental data (Supplementary Fig. 2j–m, dashed lines and dots represent model fitting and experimental data, respectively, RMSE in Supplementary Fig. 2j–m legends).

In time-course experiments, both EF1a-DDrtTA_TRE3G-mCherry and EF1a-DDrtTA_TRE3G-DDmCherry mESCs switched-off the fluorescent reporter upon inducer removal faster than cells carrying the standard Tet-On system (compare Supplementary Fig. 2n, o -dots representing data in Supplementary Fig. 2r, s- and Fig. 2d). As compared to the EF1a-rtTA_TRE3G-DDmCherry system, the DD presence on both the destabilising domain and mCherry did not further reduce the switch-off time, (compare data in Supplementary Fig. 2o and in Fig. 2e); validation of the models on time-course data (Supplementary Fig. 2n, o, dashed lines) confirmed these results. We argue that the observed switch-off dynamics are caused by a partial persistence of protein stability despite TMP removal, as also previously shown by blocking protein synthesis in EF1a-rtTA_TRE3G-DDmCherry mESCs in absence of TMP (Fig. 1g; Supplementary Fig. 1b). Furthermore, EF1a-DDrtTA_TRE3G-

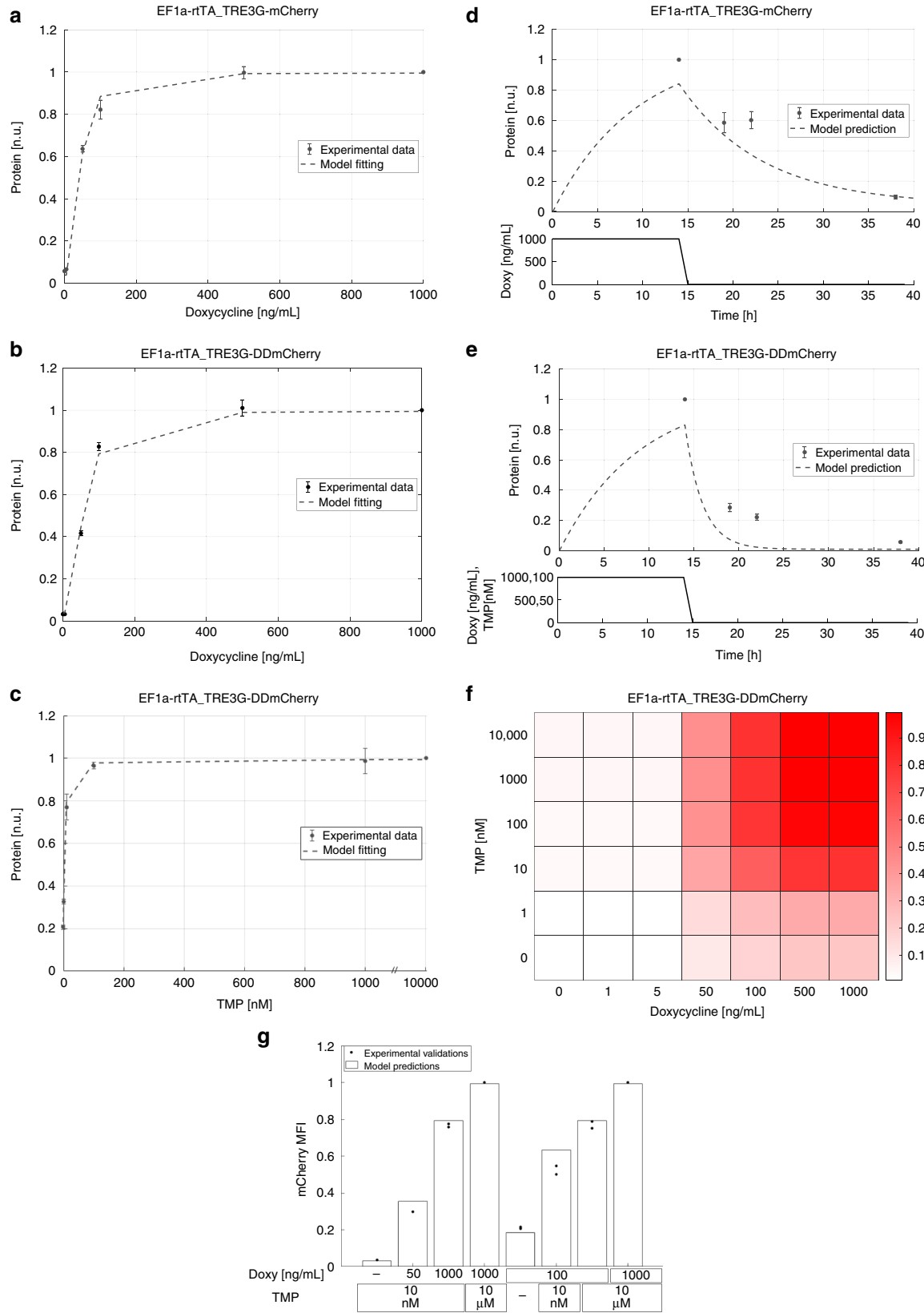

DDmCherry mESCs showed reduced activation in presence of both Doxy and TMP (compare Supplementary Fig. 2b and q), possibly due to limited TMP efficacy in stabilising both DDrtTA and DDmCherry.

Nevertheless, the dual-input control system allows fully tunable dynamic range of protein levels, as shown by dose-response model simulations (Fig. 2f; Supplementary Fig. 2t, v). We simulated steady-state cellular response to different combinations

**Fig. 2** Steady-state/dynamic response and dynamic range of the dual-input system. **a** mCherry protein levels in EF1a-rtTA_TRE3G-mCherry mESCs following 24 h of treatment with increasing concentration of Doxy (0; 1; 5; 50; 100; 500; 1000 ng/mL). **b, c** DDmCherry protein levels measured in EF1a-rtTA_TRE3G-DDmCherry following 24 h of treatment with increasing concentration of Doxy (0, 1, 5, 50, 100, 500, 1000 ng/mL) and maximum TMP (10 μM) (**b**), or increasing concentration of TMP (0, 1, 10, 100 nM; 1, 10 μM) and maximum Doxy (1000 ng/mL) (**c**). **d, e** Dynamic response of EF1a-rtTA_TRE3G-mCherry (**d**) and EF1a-rtTA_TRE3G-DDmCherry (**e**) mESCs to Doxy (1000 ng/mL) or Doxy/TMP (1000 ng/mL and 100 nM, respectively) in a time-course experiment of 38 h, in which inducers were washed out after incubation in the first 14 h. The dots are experimental data of MFI (Supplementary Fig. 2a–d) measured by flow cytometry and normalised over the maximum activation point. Dashed lines represent model fitting of steady-state data (**a–c** RMSE 0.0558 in **a**, 0.0369 in **b**, 0.0534 in **c**) or model prediction of switch-off dynamics (**d, e** RMSE 0.1590 in **d**, 0.1773 in **e**). **f** Simulated steady-state of EF1a-rtTA_TRE3G-DDmCherry mESCs upon combined treatment with Doxy and TMP at various concentrations. DDmCherry simulated values are shown as heatmaps of scaled values across the entire dynamic range of expression levels. **g** Experimental validation (dots) of simulations (bars) in (**f**), measuring EF1a-rtTA_TRE3G-DDmCherry mESC steady-state following 24 h induction with constant concentration of Doxy (100 ng/mL) and varying TMP (0, 10 nM; 10 μM), or constant TMP (10 nM) and varying Doxy (0, 50, 1000 ng/mL). Maximum concentrations of Doxy and TMP (1000 mg/mL and 10 μM, respectively) are used as control. Data are means ± SD (n = 3, **a–c**); ±SEM (n = 3 (**d, e**); n = 2, (**g**)). Source data are provided as a Source Data file

of Doxy and TMP in EF1a-rtTA_TRE3G-DDmCherry, EF1a-DDrtTA_TRE3G-mCherry and EF1a-DDrtTA_TRE3G-DDmCherry mESCs and found a robust dose-response increase in mCherry protein levels (Fig. 2f; Supplementary Fig. 2t, v). Experimental validation of model simulations (Fig. 2g; Supplementary Fig. 2u, w) confirmed both validity of the models, and the ability of tightly controlling the exogenous protein dynamic range when modulating both drugs.

We also proved the functionality of our dual-input system in a different mammalian cell line. We stably transduced HeLa cells with mCherry (henceforth HeLa_EF1a-rtTA_TRE3G-mCherry) or the conditionally destabilised DDmCherry (henceforth HeLa_EF1a-rtTA_TRE3G-DDmCherry) inducible constructs. HeLa cells showed good dose-response in steady-state activation (Supplementary Fig. 2x); the overall reporter intensity was higher than in mESCs (compare Supplementary Fig. 2b with 2x), possibly due to different transduction efficiency and/or to cell line-specific transcription, translation and degradation machinery. From titration experiments, we selected Doxy100ng/mL and TMP 10 nM to run a switch-off time-course. As in mESCs, protein stability control enabled faster On-Off transition: HeLa_EF1a-rtTA_TRE3G-DDmCherry cells showed 50% protein reduction 8 h after inducer removal, while HeLa_EF1a-rtTA_TRE3GmCherry cells still presented 70% of maximal mCherry expression after 24 h from inducer removal (Supplementary Fig. 2y, z). We observed an increase in mCherry expression in HeLa_EF1a-rtTA_TRE3G-mCherry after inducer withdrawal (Supplementary Fig. 2y), not notable in EF1a-rtTA_TRE3G-mCherry mESCs (Supplementary Fig. 2c) or HeLa_EF1a-rtTA_TRE3G-DDmCherry (Supplementary Fig. 2z); this might be due to cell type-specific Doxy metabolism (e.g. uptake, clearance), compensated by the DD presence. These data further confirm the suitability of the dual-input control system to achieve robust modulation of protein levels across cell species.

**In silico feedback control using dual-input regulation**. Recently, successful attempts have been made to dynamically regulate gene expression patterns in living cells, applying control engineering paradigms and using microfluidics or optogenetics platforms for dynamic cell stimulation, and microscopy or flow cytometry for obtaining real-time cell read-outs.[24–33] Applications in mammalian cells have been recently attempted[25], controlling a Tet-Off promoter-driven fluorescent protein using tetracycline as control input; this pioneering work showed feasibility of the in silico feedback control action, but highlighted challenges in controlling a system with slow kinetic response.

We wondered if our dual-input system would allow finer in silico gene expression feedback control using a microfluidic/microscopy platform (Supplementary Fig. 3); thus, we tested the

effectiveness of individual gene transcription or protein stability regulation as compared to their combined modulation. EF1a-rtTA_TRE3G-DDmCherry mESCs, Doxy/TMP treated, showed the same activation profile of EF1a-rtTA_TRE3G-Cherry mESCs (Supplementary Fig. 2a, b, Doxy and Doxy/TMP respectively); thus, in silico feedback control experiments were performed on EF1a-rtTA_TRE3G-DDmCherry mESCs only using different combination of control inputs. Specifically, for set-point control experiments (Supplementary Fig. 3), we applied a Relay control strategy to reduce protein expression to 50% of the maximal induction reached upon overnight treatment with combined Doxy and TMP (set-point control reference, Supplementary Note 2)[25]. Control inputs were: time-varying Doxy (Fig. 3a; Supplementary Movie 1), TMP (Fig. 3b; Supplementary Movie 2), and Doxy/TMP (Fig. 3c; Supplementary Movie 3). We found that the control action fails in controlling DDmCherry levels to reach and maintain the set-point when only modulating gene transcription (Fig. 3a; Supplementary Movie 1; Supplementary Table 3 and Note 2) or protein stability (Fig. 3b; Supplementary Movie 2; Supplementary Table 3 and Note 2), whereas a vastly improved control is achieved when both TMP and Doxy are applied (Fig. 3c; Supplementary Movie 3; Supplementary Table 3 and Note 2). Importantly, cells reach the set-point with faster dynamics than those previously observed using a Tet-Off system[25].

We compared the control performance modulating mCherry transcriptional rate in EF1a-rtTA_TRE3G-DDmCherry mESCs, but keeping the mCherry protein destabilised (i.e. in absence of TMP) both before and during the time-lapse control experiment (Fig. 3d; Supplementary Movie 4); we found a comparable control performance as when controlling the system with both inducers (Supplementary Table 3 and Note 2). This indicates that controlling a destabilised protein at transcriptional level only is a good strategy for set-point regulation, as previously shown[25]. Nevertheless, with a time-varying, multi set-point staircase reference (Fig. 3e, f; Supplementary Movies 5, 6) and inputs as in Fig. 3c, d experiments (i.e. Doxy/TMP for over-night activation and as control input, or Doxy for over-night activation and as control input, respectively), the control performance was superior when controlling the system at both transcriptional and post-translational levels (Supplementary Table 3 and Note 2). As aforementioned, in absence of TMP, the maximal EF1a-rtTA_TRE3G-DDmCherry mESC induction with Doxy is lower (Supplementary Fig. 2d); indeed, in microscopy/microfluidics experiments, we had to adapt settings to appreciate a good fluorescence signal, comparable to Doxy/TMP induced cells and informative for online protein quantification (Supplementary Note 2). Therefore, while the fluorescence signal is normalised to 1 for maximal induction, the actual absolute values of the multi

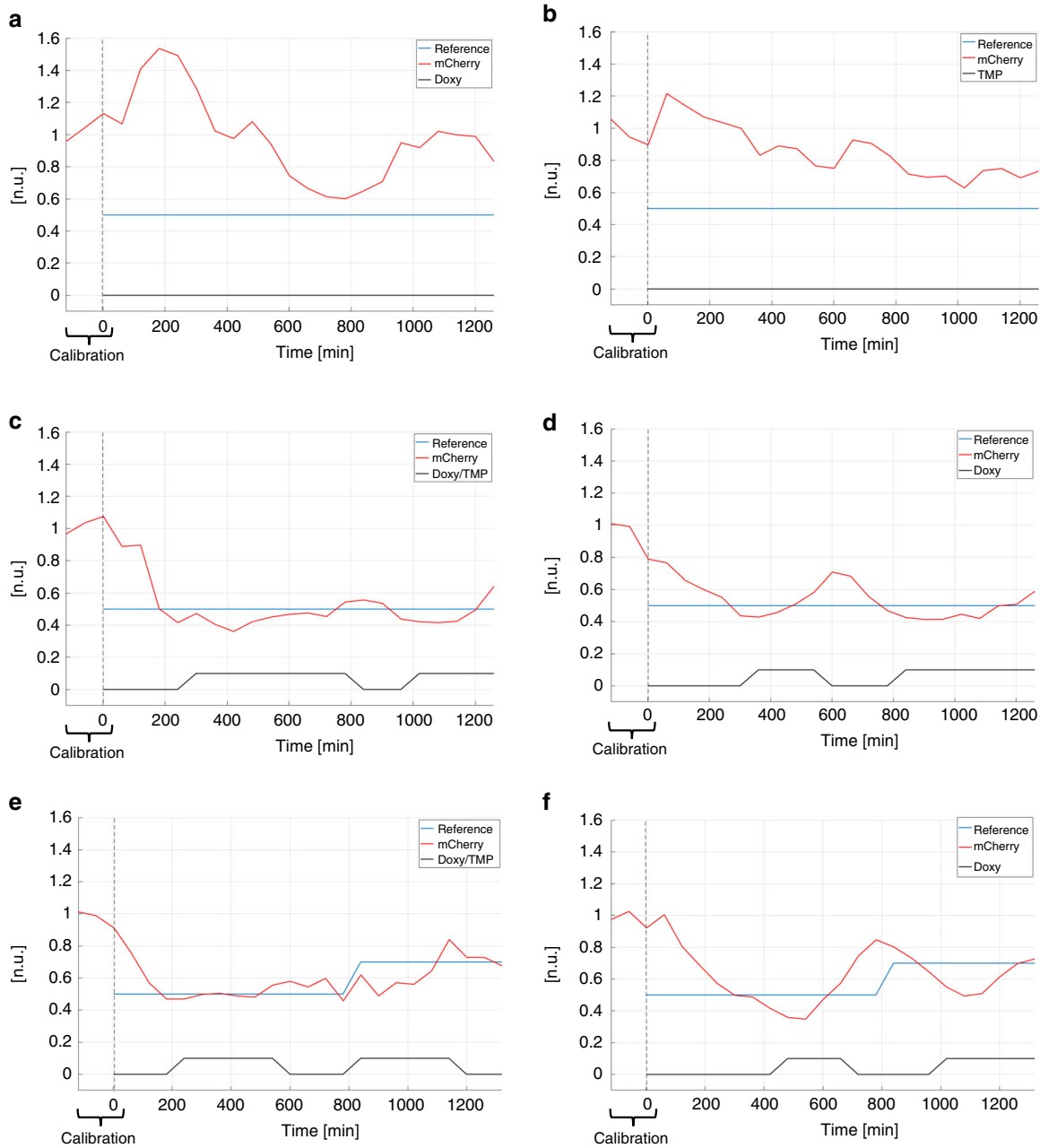

**Fig. 3** In silico feedback control of gene expression in mESCs. Set-point (**a–d**) and time-varying (**e**, **f**) reference relay control experiments, using inducers (Doxy 1000 ng/mL and/or TMP 100 nM) as control inputs. In red the measured output (normalised DDmCherry fluorescence), in blue the control reference fluorescence, set at 50% (**a–d**) or 50–70% (**e**, **f**) of the average maximal induction values measured during the calibration phase (120 mins with continuous Doxy/TMP (**a–c**, **e**) or Doxy (**d**, **f**) administration). The time-lapse sampling time was set at 60 mins. The effectiveness of controlling transcription (**a**, **d**, **f**), protein stability (**b**) and both (**c**, **e**) was tested: cells received either media with Doxy (**a**, **d**, **f**) or Doxy/TMP (**c**, **e**) or TMP (**b**) when the measured fluorescence was below the control reference, and plain media (**c–f**) or TMP (**a**) or Doxy (**b**) supplemented media when the measured fluorescence was above the control reference. The control performance indexes for each control experiment are reported in Supplementary Table 3 and Note 2. In the experiments in **d**, **f**, cells were never pre-treated with the protein stabiliser TMP (i.e. they only received Doxy). Details about feedback control experiments are in Supplementary Note 2

set-point reference signal are lower for the control experiments in Fig. 3d, f as compared to those in Fig. 3c, e. This might explain the different slope in the dynamic response to Doxy addition/removal, resulting in more evident oscillations around the set-points, which are anyway inevitable using a Relay control strategy[25] and a relatively long sampling time (60 min to avoid phototoxicity, Supplementary Note 2).

These results demonstrate that our dual-input control is preferable for Relay control-based in silico feedback regulation of

gene expression in mammalian cells, and is likely to be suitable for generating more complex time varying gene-expression dynamics.

**Dual-input control of signalling pathway in mESCs.** We then tested the general applicability our dual-input system for fine-tuning of signalling pathways. We chose the canonical Wnt signalling pathway, as its activation levels and spatiotemporal dynamics are crucial for several biological processes[34,35],

including intestinal crypt homoeostasis[36–38], somite development during embryogenesis[39], stem cell pluripotency maintenance and somatic cell reprogramming[40–42].

To test the ability of our system to modulate levels of β-catenin, the main effector of the canonical Wnt pathway[35,43], we generated mESCs stably expressing a fusion protein comprising the destabilising domain (DD), the mCherry fluorescent protein and a constitutively active β-catenin (β-catenin$^{S33Y}$)[44], driven by a doxycycline-inducible promoter (henceforth EF1a-rtTA_TRE3G-DDmCherryβ-catenin$^{S33Y}$ mESCs, Fig. 4a), in a wild-type mESC line. As with previous constructs, we found selective response of exogenous DDmCherryβ-catenin$^{S33Y}$ mRNA (Supplementary Fig. 4a, b) and protein (Fig. 4a–d; Supplementary Fig. 4c, d) levels to Doxy and Doxy/TMP treatment, respectively. To prove functionality of the recombinant protein, we measured the subcellular localisation of both endogenous and exogenous forms of β-catenin and found comparable patterns upon inducer treatment (Fig. 4c, d; Supplementary Fig. 4c, d). Furthermore, nuclear translocation of the exogenous protein was detected from western blot on subcellular fractions (Supplementary Fig. 4e), confirming full functionality of the conditional protein.

Next, we tested DDmCherryβ-catenin$^{S33Y}$ tunability by running in silico feedback control experiments, using Doxy/TMP as time-varying inputs as they allowed robust set-point regulation in EF1a-rtTA_TRE3G-DDmCherry mESCs (Fig. 3c). In presence of concentrations of Doxy (1000 ng/mL) and TMP (100 nM) used in previous feedback control experiments (Fig. 3c, e), DDmCherryβ-catenin$^{S33Y}$ protein levels were stable and never reached the desired set-point (Supplementary Fig. 4f; Supplementary Movie 7; Supplementary Table 3 and Note 2), in contrast with the experiment in Fig. 3c, likely due to different protein half-life and size of the DDmCherry as compared to the DDmCherryβ-catenin$^{S33Y}$ [41]. Therefore, we performed titration experiments of EF1a-rtTA_TRE3G-DDmCherryβ-catenin$^{S33Y}$ mESCs (Supplementary Fig. 4g) and consequently selected lower concentrations of Doxy (100 ng/mL) and TMP (10 nM), still enabling full activation but possibly easier to be washed-out from cells, to run set-point control experiments. Lower concentration of inducer molecules allowed robust feedback control (Fig. 4e; Supplementary Movie 8; Supplementary Table 3 and Note 2), demonstrating that control of biologically relevant proteins is possible and that, irrespectively of half-life and protein turnover, our system can fine tune gene-of-interest levels.

To study the biological relevance of the inducible DDmCherryβ-catenin$^{S33Y}$ protein, we took advantage of a mESC line deleted for endogenous β-catenin gene we generated (β-catenin$^{-/-}$ mESCs)[45], and tested whether it was able to rescue the impaired clonogenicity and survival of β-catenin null mESCs grown in absence of the Leukaemia Inhibitory Factor (LIF)[45–47]. We stably transduced β-catenin$^{-/-}$ mESCs[45] with the EF1a-rtTA_TRE3G-DDmCherryβ-catenin$^{S33Y}$ plasmids (henceforth C1-EF1a-rtTA_TRE3G-DDmCherryβ-catenin$^{S33Y}$ mESCs). As with previous constructs, we found selective dose-response of exogenous DDmCherryβ-catenin$^{S33Y}$ mRNA (Supplementary Fig. 4h, i) and protein (Supplementary Fig. 4j–l) levels to Doxy and Doxy/TMP treatment, respectively. We also measured the levels of total, cytosolic and nuclear DDmCherryβ-catenin$^{S33Y}$ protein upon induction by western-blot (Supplementary Fig. 4m). All extracts showed an increase in the amount of DDmCherryβ-catenin$^{S33Y}$, when cells were Doxy/TMP treated (Supplementary Fig. 4m), consistent with flow cytometry data (Supplementary Fig. 4l). As compared to β-catenin amount in wild-type cells, only in nuclear extracts inducible DDmCherryβ-catenin$^{S33Y}$ levels are above the endogenous ones (Supplementary Fig. 4m, Nuclear); this observation is coherent with the constitutive active form of

the protein we used, which is insensitive to the endogenous degradation machinery and therefore accumulates into the nucleus more than the native β-catenin.

To investigate the effect of DDmCherryβ-catenin$^{S33Y}$ over-expression, we performed a clonogenicity assay in 5 different culture conditions following LIF withdrawal. C1-EF1a-rtTA_TRE3G-DDmCherryβ-catenin$^{S33Y}$ mESCs were exposed to TMP 10 μM and increasing concentrations of Doxy (10–100 ng/mL; Supplementary Fig. 4n), and pluripotent colonies were detected by Alkaline Phosphatase (AP), which only stains cells retaining the pluripotent phenotype[48] (Methods). When uninduced cells were cultured in serum- and LIF-free media, as previously reported[46,47], the number of AP positive cells was null or the signal was faint, while in NDiff-Chiron/PD, NDiff-Chiron and NDiff-PD media supplemented with TMP10 μM and Doxy10 ng/mL, but not Doxy 100 ng/mL, some AP + colonies appeared, rescuing the impaired phenotype (Supplementary Fig. 4o). These results, while confirming the previous observation of LIF dependency in β-catenin-deficient mESCs[46,47], indicate that levels of exogenous β-catenin need to be kept within a certain threshold for rescuing pluripotency, as excessive amounts most likely could instead promote cells differentiation[42].

Finally, to demonstrate the general applicability of our dual-input system, we applied it to control two proteins not tagged with fluorescent reporters, specifically β-catenin$^{S33Y}$ and its transcriptional partner Lymphoid enhancer binding factor1 (Lef1)[49,50]. We generated mESCs stably expressing the transactivator and a conditionally destabilised β-catenin$^{S33Y}$ (DDβ-catenin$^{S33Y}$) protein in β-catenin$^{-/-}$ mESCs (henceforth EF1a-rtTA_TRE3G-DDβ-catenin$^{S33Y}$, Fig. 4f) or the conditionally destabilised Lef1 (DDLef1) protein in wild-type mESCs (henceforth EF1a-rtTA_TRE3G-DDLef1, Fig. 4g). Consistently with our previous observations, DDβ-catenin$^{S33Y}$ (Supplementary Fig. 4p) and DDLef1 (Supplementary Fig. 4q) mRNA got activated upon Doxy treatment and did not change in presence of TMP; protein levels varied as expected at steady-state upon induction (i.e. titration experiments in Fig. 4h, i) and in switch-off time courses (Fig. 4j, k). Immunofluorescence staining also confirmed adequate DDβ-catenin$^{S33Y}$ (Supplementary Fig. 4r, s) and DDLef1 (Supplementary Fig. 4t, u) protein induction and correct subcellular localisation.

These findings open potential avenues for dynamic control of protein expression in stem cells.

## Discussion

Complex time-varying patterns of gene expression underlie numerous biological processes such as immune regulation and cell fate choice[51,52]; mimicking and perturbing these profiles of expression requires tools that faithfully recapitulate often fluctuating kinetics.

In this study, we presented an enhanced tool for conditional gene expression regulation in mammalian cells encompassing a Tet-On 3G system combined with DD technology that allows fast, specific, reversible and tunable perturbation of biological macromolecules. While these and related systems had already been proposed in the literature, their combination had not been fully explored previously.

We demonstrated experimentally and with a computational model that by incorporating control at the transcriptional level, as well as regulating protein-of-interest stability, allows a much finer and tighter control of gene expression kinetics as compared to transcriptional or post-translational control alone. Indeed, in our system genes of interest can be robustly induced but also more rapidly switched off, likely by enhanced proteasomal degradation of the translated protein, compared to either stand-alone transcriptional or post-translational control. While it is true that

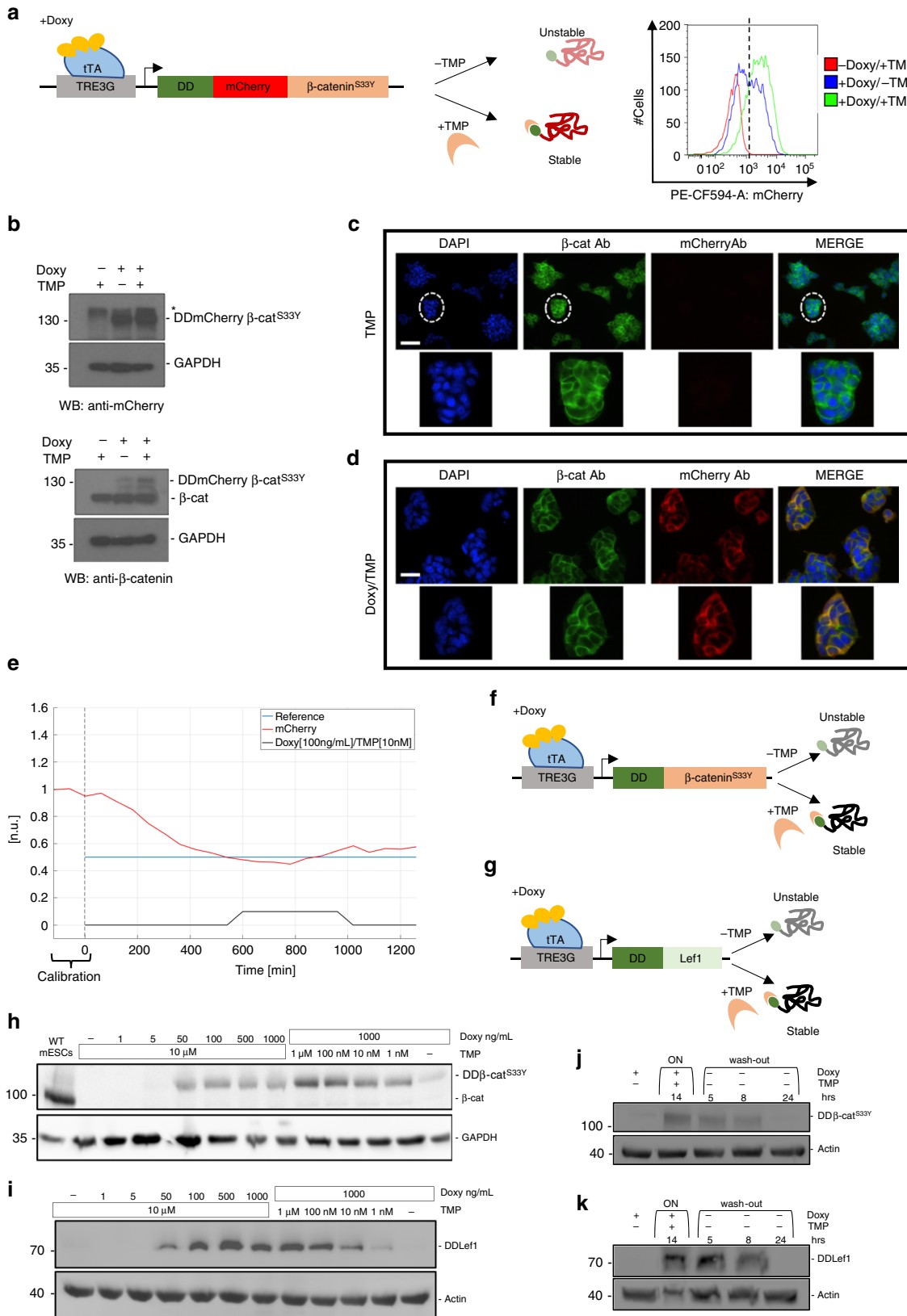

inducible transcription of a constitutively destabilised protein could be also used (e.g. fused to Ubiquitin)[53], for some proteins maximising constitutively the degradation kinetics might impair the relative biological function; furthermore, the maximal expression level and, consequently, the dynamic range of the

protein will also be diminished, as we showed when treating EF1a-rtTA_TRE3G-DDmCherry mESCs with Doxy only (Supplementary Fig. 2d).

These results are in line with the different dynamic range of a Tet-On/Off system obtained fusing various degron tags to rtTA/

**Fig. 4** Dual-input control of β-catenin and Lef1 levels in mESCs. **a**, left panel, **f**, **g** Dual-input regulation system consisting of the reverse transactivator (rtTA) and the conditionally destabilised β-catenin$^{S33Y}$ (tagged or not with mCherry, **a** left panel and **f**, respectively), or Lef1 (**g**). **a**, right panel Representative flow cytometry profile of Doxy and/or TMP treated EF1a-rtTA_TRE3G-DDmCherryβ-catenin$^{S33Y}$ mESCs. **b** DDmCherryβ-catenin$^{S33Y}$ protein levels measured by western blot in EF1a-rtTA_TRE3G-DDmCherryβ-catenin$^{S33Y}$ mESCs treated with Doxy and/or TMP; mCherry (upper panel) and β-catenin (lower panel) antibodies were used. Asterisk indicates a non-specific band. **c**, **d** β-catenin immunostaining in EF1a-rtTA_TRE3G-DDmCherryβ-catenin$^{S33Y}$ mESCs treated for 24 h with TMP (**c**) or Doxy/TMP (**d**), using mCherry (red signal) and β-catenin (green signal) antibodies. DAPI was used to stain the nuclei. Zoomed pictures of selected clones are shown. Scale bars 25 μm. **a–d** Doxy and TMP were used at 1000 ng/mL and 100 nM, respectively. **e** Set-point relay control experiment, using inducers (Doxy 100 ng/mL and TMP 10 nM) as control inputs. In red the measured output (normalised mCherry fluorescence), in blue the reference fluorescence, set at 50% of the average values measured during the calibration phase (120 mins with continuous Doxy/TMP administration). The control performance indexes are reported in Supplementary Table 3 and Note 2; details about feedback control implementation are in Supplementary Note 2. **h**, **i** DDβ-catenin$^{S33Y}$ (**h**) and DDLef1 (**i**) protein levels measured by western blot in EF1a-rtTA_TRE3G-DDβ-catenin$^{S33Y}$ (**h**) and EF1a-rtTA_TRE3G-DDLef1 (**i**) mESCs following 24 h of treatment with increasing concentration of Doxy (0, 1, 5, 50, 100, 500, 1000 ng/mL) and maximum TMP (10μM), or increasing concentration of TMP (0, 1, 10, 100 nM; 1, 10 μM) and maximum Doxy (1000 ng/mL). Samples induced with the maximal concentration of inducers (Doxy1000ng/mL and TMP10μM) were loaded only once. **j**, **k** Dynamic response of EF1a-rtTA_TRE3G-DDβ-catenin$^{S33Y}$ (**j**) and EF1a-rtTA_TRE3G-DDLef1 (**k**) mESCs to Doxy/TMP in a time-course experiment of 38 h, in which inducers were washed out after 14 h incubation. The concentration of inducer molecules used are Doxy 100 ng/mL and TMP 10 nM (**j**) or Doxy 1000 ng/mL and TMP 100 nM (**k**). Source data are provided as a Source Data file

tTA, and measuring the response of a target gene driven by a TRE promoter[54]. Our system, thanks to conditional protein regulation, allows modulation of protein stability and, consequently, of the system dynamic range and On-Off dynamics without requiring the engineering of different constructs and cell lines.

We showed reduced cell-to-cell variability in single cell induced response when controlling both transcription and protein stability, as compared to a Tet-on system; in light of recent results in mammalian cells indicating strong variability in protein degradation rates in isogenic and phenotypically homogeneous populations, with consequent effect on extrinsic noise[55], we suggest that conditional protein regulation could be further tested for noise modulation. Single-cell experiments comparing how conditional regulation of mRNA and/or protein dynamics can affect both intrinsic and extrinsic noise[56] might be interesting avenues for future research into origins and tunabilty of cellular noise, and its effects on cellular phenotypes.

Inducible promoters enable transcriptional regulation and, combined with systems for controlled protein levels modulation, provide valuable tools for controlling the expression rate and amount of proteins of interest. The plant derived AID (auxin-inducible degradation) system has been recently used to get fast and efficient proteasomal degradation of AID-tagged proteins in response to auxin hormones[57]. However, rates of AID-mediated protein degradation/recovery strongly depend not only on auxin uptake and metabolism, but also on the abundance of SCF complex components, which might vary among different biological systems[58,59]. Therefore, we believe that the AID system is more suitable for protein knockout experiments instead of fine temporal control of fused-protein levels. Of note, in yeast and humans it has also been noted a reduction in the amount of AID-fused proteins in absence of auxin, suggesting premature protein degradation in cells expressing the auxin responsive F-box protein, Tir1[60,61]. We tested AID-mediated protein degradation by stably introducing in mESCs a plasmid containing both the OsTir1 (CAG-OsTir1-V5–2A-PuroR) and a miniAID-fused mCherry (EF1a-mCherrymAID; Supplementary Fig. 5a) in combination with a Cas9 expression vector for single copy insertion[57,60,62]. Although drug-selected, the percentage of mCherry expressing cells were barely detectable by flow cytometry (Supplementary Fig. 5b) and, when analysed by confocal microscopy, showed aberrant patterning of the fusion protein (Supplementary Fig. 5c). We confirmed correct expression of the OsTir1 by western-blot but failed in detecting any mCherry signal (Supplementary Fig. 5d). These results are in agreement with previous findings about premature AID-fused protein

degradation in absence of the auxin analog Indole-3-acetic acid sodium salt (IAA)[60,61]. To exclude the observed phenotype was cell-specific, we carried out a transient experiment in HEK293T cells. We transfected cells with untagged mCherry- or mCherrymAID-carrying plasmids, and monitored fluorescent protein expression and distribution by western-blot and confocal microscopy. Two days after transfection, mCherrymAID levels were almost undetectable as compared to cells carrying the untagged mCherry (Supplementary Fig. 5e, f); even when increasing laser settings, we could only detect small protein aggregates (Supplementary Fig. 5f) and mCherrymAID levels were much lower than those of the untagged mCherry (Supplementary Fig. 5g), indicating basal protein degradation in absence of IAA. All together, these results showed the AID system has some limitations impeding further experiments of dynamic protein control, and tagging of endogenous proteins for which localisation is important (e.g. β-catenin). In contrast, the destabilisation domain we applied in this study shows good stability in presence of the inducer molecule TMP, and requires minimal genetic manipulations of the recipient cellular system[58,59]. Furthermore, as the inducer stabilises the protein, the dynamic range of the inducible system can be fully controlled to obtain 'on-demand' activation levels.

We demonstrated that our dual-input regulation system is suitable for microfluidics/microscopy-based in silico feedback control: it outperforms transcriptional- and post-translational-only manipulation to reach and maintain both set-point and time-varying multi set-point control references, even using a simple and model-free control strategy. Using more advanced control strategies, such as Model Predictive Control[63], might overcome limitations of transcriptional- or post-translational-only regulation, and avoid oscillations that we and others[25] reported when using a Relay strategy; nevertheless, such controllers require a mathematical model of the system, not always available.

Our system suits generation of temporal patterns of gene expression that traditionally have not been easy to manipulate. For instance, we demonstrated that finely tuned levels of Wnt/β-catenin pathway genes (i.e. β-catenin and Lef1) are achievable, despite the half-life of the tagged proteins, and can exert specific functions on mESC pluripotency. The canonical Wnt pathway is known to display oscillating patterns of gene expression in stem cells and in the developing embryo[39–41], which are a key determinants of cell fate determination. This suggests that more complex dynamic patterns of Wnt-pathway activation could be engineered; mimicking in silico native temporal patterns should

be compatible with the On-Off kinetics we measured, given the relatively slow period and amplitude of oscillations observed in vivo[39]. Our platform, which has the potential to allow a quantitative assessment of the molecular mechanisms and dynamics underpinning cell fate choices, paves the way for controlling cellular behaviour in clinically relevant model systems, such as stem cells.

The inducibility of our system makes it a powerful perturbation method, and the ability to tune protein levels with a high degree of predictability (as evidenced by our faithful capture of system dynamics using a computational model) makes it a valuable resource with a broad scope of applications. For instance, it could be harnessed to study how signalling networks or synthetic gene circuits are wired together, by fine manipulation of one 'node' in the network and observing how this affects other cognate members within the circuit. Ultimately, this approach could be used to deconvolute and quantitatively assess the interactions within these networks, which could both explain and predict the consequences of a given perturbation. The mathematical model we fitted and validated could be used to test the design of new synthetic networks based on inducible promoters and destabilising domains[23,64,65].

Finally, the ability of both Doxy and TMP to cross the placental barrier[66] opens up the possibility that our dual-input system could allow fine-tuning of genes and signalling pathways essential in embryonic development, and could be applicable as a novel approach for targeted gene therapy.

## Methods

**Inducible constructs**. pLVX_EF1a-Tet3G Neo (in the manuscript referred as EF1a-rtTA) and pLVX_TRE3G Puro were purchased from Clontech (631363). DHFR-derived destabilization domain (DD), mCherry and β-catenin$^{S33Y}$ were PCR amplified from pOddKS[67], 7TGC[68] (Plasmid #24304) and pMXs-beta-catenin-S33Y (Plasmid #13371) plasmids, respectively. Lef1 cDNA was amplified from total RNA extracted from mouse embryonic stem cells using SuperScript III first strand cDNA synthesis kit. The reverse transcriptase reaction was primed with a reverse oligo annealing to the 3′ end of Lef1 CDS (5′-TCAGATGTAGGCAG CTGTCA-3′). Five microlitre of the reverse transcription reaction were used to amplify Lef1 using the following oligos: For: 5′-ATGCCCCAACTTTCCGGAG G-3′ Rev- 5′-TCAGATGTAGGCAGCTGTCA −3′). Following manufacturer's instructions, fragments were assembled into pLVX_TRE3G Puro (Clontech) and linearised with SmaI (NEB) using HiFi-DNA assembly cloning kit (NEB). pLVX_TRE3G-DDmCherry is available on Addgene (Plasmid #108679). EF1a mCherrymAID fusion was cloned into pEN396 (Addgene #92142) linearised with NotI through Gibson Assembly to generate the all-in-one vector expressing CAG-Tir1 and EF1a-mCherrymAID fusion protein. mAID was amplified from pMK289 (Addgene #72827).

**Cell line derivation**. Stable cell lines were generated by lentiviral infection of R1 (EF1a-rtTA_TRE3G-mCherry, EF1a-rtTA_TRE3G-DDmCherry, EF1a-DDrtTA_-TRE3G-mCherry, EF1a-DDrtTA_TRE3G-DDmCherry and EF1a-rtTA_TRE3G-DDLef1), E14Tg2a (EF1a-rtTA_TRE3G-DDmCherryβ-catenin$^{S33Y}$) and β-catenin−/−[45] (EF1a-rtTA_TRE3G-DDmCherryβ-catenin$^{S33Y}$ and EF1a-rtTA_TRE3G-DDβ-catenin$^{S33Y}$) mouse embryonic stem cells (mESCs) adapting the protocol used in ref. [69]. Briefly, $7 \times 10^5$ HEK293T cells were seeded in a 6-well plate 24 h before the transfection in Dulbecco's modified Eagle's medium supplied with 10% FBS (Life Technologies), 10 U/mL penicillin, 10 μg/mL streptomycin, 2 mM glutamine, 1 mM sodium pyruvate, and 1x nonessential amino acids. The day after cells were co-transfected with 1 μg of each specific expression vector, 0.75 μg of psPAX.2, and 0.25 μg of pMD2.9 packaging plasmids. Twenty-four h after transfection, cells were switched from HEK293T to mESCs growth medium, and the media was collected after additional 24 and 48 h and used to infect recipient mESCs ($5 \times 10^4$ seeded 24 h before the first infection in a 6-well plate). mESCs were firstly infected with the stable (EF1a-rtTA) or conditionally destabilised (EF1a-DDrtTA) transactivator containing vector. After Neomycin selection, cells were subjected to a second round of infection with the doxycycline-inducible vector (pLVX_TRE3GmCherry, DDmCherry, DDmCherryBcatS33Y, DDBcatS33Y or DDLef1) and Puromycin selected. Finally, for fluorescence-tagged plasmids only, to enrich for mCherry expressing cells and to homogenise mCherry levels across cell lines, we sorted mCherry positive cells after Doxy or Doxy/TMP (1000 ng/mL and 10 μM, respectively) 24 h treatment. mESCs were cultured on gelatin coated dishes in knockout Dulbecco's modified Eagle's medium (DMEM) supplemented with 20% fetal bovine serum (Sigma), 1 x nonessential amino acids, 1 x GlutaMax, 1 x2-mercaptoethanol and 1000 U/mL LIF (Peprotech).

For the AID system experiments, R1 mESCs were co-transfected with the all in-on-one vector expressing the CAG-OsTir1- V5-2A-PuroR and the EF1a-mCherrymAID fusion protein, and the Cas9 expression vector for targeted single copy integration in the TIGRE locus. Cells were puromycin selected for 3 weeks and expanded. HEK293T cells were transfected with the all in-on-one vector expressing the CAG-OsTir1- V5-2A-PuroR and the EF1a-mCherrymAID fusion protein or with the CAG-mCherry plasmid, and analysed 2 days upon transfection. Wild-type mESCs used in this study were kindly provided by Prof Austin Smith, β-catenin$^{-/-}$ mESCs were derived in the laboratory of Dr Maria Pia Cosma[45], HeLa (HeLa CCL-2) and HEK293T (CRL-11268) cells were directly purchased from ATCC. Cell lines were tested for mycoplasma.

**Flow activated cell sorting (FACS)**. Cells were washed with sterile phosphate-buffered saline (PBS, Gibco), trypsinised for 2–3′ at room temperature and centrifuged at $1000 \times g$ for 5′. Pelleted cells were resuspended in 500 μL of complete mESCs media supplemented with DAPI. The mCherry positive fraction was sorted from DAPI negative using the BD Influx high-speed 16-parameter fluorescence activated cell sorter; the gating strategy was defined to get comparable mCherry intensity across cell lines. For single cells sorting, cells were prepared as previously and individually sorted in one well of a 96-wellplate in standard culture medium.

**Flow cytometry analysis**. Cells from a 24-well plate were washed with sterile Phosphate-Buffered Saline (PBS, Gibco), incubated with 50 μL of trypsin for 2–3′ at room temperature and collected with 150 μL of PBS 2% FBS containing DAPI as cell viability marker. Cell suspension was analysed using the BD LSR Fortessa and 10,000 living cells were recorded for each sample. Both % of mCherry positive cells and Median Fluorescence Intensity (MFI) were calculated over living cells, gated as DAPI negative.

**Drug treatments**. Drugs used in this study are doxycycline (Sigma, D9891-1G), TMP (Sigma, T7883), cycloheximide (Sigma, C4859) and MG132 (Tocris, 1748). Concentrations and time of treatment are specified in main text and figure legends.

**qPCR**. For quantitative PCR, the total RNA was extracted from cells using the RNeasy kit (Qiagen), and the cDNA was generated from 1 μg of RNA. Twenty-five ng of cDNA were used as template for each qPCR reaction, in a 15 μL reaction volume. iTaq Universal SYBR Green Supermix (1725120, Bio-Rad) was used with Qiagen Rotor-Gene System. The primers used were: Actin-Fwd: ACGTTGACAT CCGTAAAGACCT, Actin-Rev: GCAGTAATCTCCTTCTGCATCC; mCherry-Fwd: GAACGGCCACGAGTTCGAGA, mCherry-Rev: CTTGGAGCCGTACATG AACTGAGG; β-catenin-Fwd: CGACACTGCATAATCTCCTGCTCC, β-catenin-Rev: GGTCCACCACTGGCCAGAATGAT; Lef1-Fwd: CCCACACGGACAGTGA CCTA, Lef1-Rev: TGGGCTCCTGCTCCTTTCT.

**Biochemical analysis**. mESC cell lines were treated as detailed in the text and figure legends; Doxy and TMP are specified in the latter; cycloheximide was used at 25 μg/mL. Cells were lysed in lysis buffer (1% TX-100, 150 mM NaCl, 15 mM Tris pH 7.5 and protease inhibitors) before processing by SDS-PAGE and western-blotting. For proteasomal inhibition experiments, MG132 (5 μM) was added 6 h prior to lysis. Antibodies used are anti-mCherry (AB0040-200, Sicgen, 1:10,000), anti-β-catenin, clone 14 (610153, BD Biosciences, 1:5000) anti-GAPDH, clone 6C5 (ab8245, Abcam, 1:10,000), anti-Tubulin, clone YOL1/34 (sc-53030, Santa Cruz, 1:2000), anti-Actin (ab8226, Abcam, 1:2000), anti V5, clone D3H8Q (13202, Cell Signaling, 1:1000), anti Lef1, clone C12A5 (2230, Cell Signaling, 1:2000), anti HDAC3 (ab7030, abcam, 1:000). Image densitometry was carried out using ImageJ.

**Ubiquitination analysis**. Cells were drug treated as specified in the text and figure legends, with Doxy (1000 ng/mL) and TMP (100 nM) for 24 h and chased for 12 h with or without TMP in the presence of MG132. Cells were then lysed in denaturing lysis buffer (2% SDS, 0.5 % NP40, 15 mM Tris pH 6.8, 5% glycerol, 150 mM NaCl) to prevent deubiquitination, before dilution in modified RIPA buffer (25 mM Tris pH7.2, 150 mM NaCl, 1% NP40, 0.5 % Sodium Deoxycholate, 1 mM EDTA) to dilute the SDS to 0.1%. Samples were then immunoprecipitated with 2 μg of anti-ubiquitin antibody (P4D1, Cell Signaling) for 4 h before addition of Protein G for a further 1 h, followed by extensive washing and processing via SDS-PAGE and western-blotting with anti-mCherry antibody (AB0040-200, Sicgen).

**Immunostaining**. Cells were washed twice with PBS, fixed with 4% paraformaldehyde for 15 mins at RT, and blocked in PBS with 0.1% Triton X-100 and 3% BSA, for 60 mins. Cells were incubated overnight with the specific primary antibody in PBS with 0.1% Triton X-100 and 1% BSA.

Cells where PBS washed for three times (5′ each wash) before being incubated with the secondary antibody conjugated with fluorescein for 1 h at room temperature. Finally, cells were washed and mounted on slides with a few drops of Vectashield, with DAPI (Vector Laboratories). The primary antibodies used for immunostaining are: anti-β-catenin, clone 14 (610153, BD Biosciences), working concentration 1:1000; anti-mCherry (AB0040-200, Sicgen) working concentration

1:1000; anti Lef1, clone C12A5 (2230, Cell Signaling, 1:1000). The fluorescent conjugated secondary antibodies were diluted 1:1000.

**Nucleus/cytosol fractionation**. For Nuclear and Cytosolic protein fractionation, cells from a 10 cm dish were washed in cold PBS, scraped in 1 mL of cold PBS using a cell scraper and centrifuged (5 min at $12,000 \times g$). The supernatant was discarded, and the pellet resuspended in 900 μL of cold 0.1% NP40 in PBS. Three hundred microlitre of the suspension was remove as total extract and the remaining 600 μL centrifuged for 5 min at $12,000 \times g$. After the centrifugation, 300 μL of supernatant was removed as cytosolic fraction and the remaining 300 μl discarded. The nuclear pellet was washed with 1 mL of cold PBS, centrifuged as above, the supernatant discarded, and the pellet resuspended in 180 μL of 1x Laemmli buffer. Hundred microlitre of 4x Laemmli buffer was added to both total and cytosolic fractions. Samples were sonicated (level 2 thrice for 10 s each) and boiled for 3 min before being loaded (20 μL of total and cytosolic and 10 μL on nuclear extracts).

**Alkaline phosphatase (AP) assay**. At Day1 C1-EF1a-rtTA_TRE3G-DDmCherryβ-catenin$^{S33Y}$ mESCs were plated in 12-well plate to a density of 131cells/cm$^2$ and cultured for 24 h in standard (i.e. serum enriched) mESC growth media without LIF. At Day2 cells were switched to different media and cultured for around 1week always in absence of LIF and with the addition of the inducers TMP+/−Doxy (10 μM and 10–100 ng/mL, respectively). The serum-free media are all based on NDiff227[70] (Y40002, Takara) medium supplemented or not with the Mek (PD; PD0325901, S1036 Selleckchem) and/or the GSK3 (Chiron; CHIR-99021, S1263 Selleckchem) inhibitors[70]. At Day 8 from the plating cells were fixed in 10% cold Neutral Formalin Buffer (NFB) for 15 min at 4 °C followed by 15 min in distilled water and incubated for 45–60 min (until red colonies are visible) with the Alkaline Phosphatase substrate containing Naphthol (N4875, Sigma), N,N-Dimethylformamide (DMF, 227056, Sigma), 0.2 M Tris-HCl and Fast Red Violet LB Salt (F3381, Sigma).

**Single clone coefficient of variation calculation**. As a measure of the mCherry variation across clones, we calculated the coefficient of variation (SD divided by the mean value) on the mCherry+ cells for both EF1a-rtTA_TRE3G-mCherry and EF1a-rtTA_TRE3G-DDmCherry single clones. The coefficient of variation (CV) indicates how much the single clones deviate from the average. The Higher the CV value is, the more heterogeneous the cellular response is.

**Statistical analysis**. Differences between groups were analysed by Student $t$ test and nonparametric one-way ANOVA using GraphPad Prism (version 8.2.0). A $p$-value lower than 0.05 was considered statistically significant.

## Data availability
All data presented in this study are available from the corresponding authors upon reasonable request. Source data are available in the Source Data file.

## Code availability
All code used in this study is available from the corresponding authors upon reasonable request.

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

## Acknowledgements

We thank Prof Austin Smith for the wild-type mouse Embryonic Stem Cells used to generate EF1a-rtTA_TRE3G-DDmCherryβ-catenin$^{S33Y}$ mESCs; Prof. Shinya Yamanaka for the pMXs-beta-catenin-S33Y plasmid; Dr Andre Hermann and Dr Lorena Sueiro Ballesteros (Flow Cytometry Facility, University of Bristol), and Dr Mark Jepson and Alan Leard (Wolfson Imaging Facility, University of Bristol) for their support. This work was funded by Medical Research Council grant MR/N021444/1 to L.M., by the Engineering and Physical Sciences Research Council grant EP/R041695/1 to L.M., and by BrisSynBio, a BBSRC/EPSRC Synthetic Biology Research Centre (BB/L01386X/1) to L.M. and by the Mexico Consejo Nacional de Ciencia y Tecnología (CONACyT) PhD scholarship provided to SMO.

## Author contributions

E.P. generated inducible cell lines; E.P. designed and performed experiments; L.P. implemented the control strategy; L.P., S.M.O. and L.M. developed the mathematical model; F.A. generated inducible plasmids and the β-catenin$^{-/-}$ cell line; E.P., D.L.R. and M.K. performed experiments; D.d.B. supported microfluidic-based experiments; M.P.C. supported plasmid and β-catenin$^{-/-}$ cell line generation; E.P., D.L.R., L.P. and L.M. analysed data; E.P., D.L.R., L.P. and L.M. wrote the paper; L.M. supervised the entire project.

## Competing interests

The authors declare no competing interests.
