## [Peer Review File · Nature Communications]

Reviewers' comments:

Reviewer #1 (Remarks to the Author):

The manuscript by Pedone et al. Entitled 'A tunable dual-input system for 'on-demand' dynamic gene expression regulation' describes a synthetic biology system relying on independent control inputs for gene expression and protein stability and its application to dynamic control of protein abundances in mammalian cells. Compared to control of gene expression alone, the authors argue that their system decreases response times, enhances tunability and dynamic range, and improves controllability in *in silico* feedback applications. Proof-of-principle applications include the control of signaling components in stem cell with potential future biomedical applications.

While many synthetic biology systems for gene expression control in mammalian systems have been developed, relatively few systems for the control of protein stability (e.g., via small molecules such as rapamycin) exist, and the combination of both approaches with orthogonal controllability is novel. A strength of the manuscript is the careful experimental characterization of the system combined with mathematical modeling. The proof-of-principle applications of *in silico* feedback also demonstrate feasibility in the complex stem cell setting.

In several aspects, however, improvements of the manuscript appear warranted, primarily in clarifying the advances over the state of the art, support for some claims, and application perspectives. More specifically:

(i) The basic design of the system relies on previously established and characterized components (tTA and DD-fusions), such that it is unclear to what extent the basic experimental characterization (Figs. 1,2) reveals novel (unexpected) features. A more detailed discussion / comparison with features of the original (related) control systems should be provided, for example, regarding robustness, specificity, precision, mechanisms of proteasomal degradation, steady-state characteristics.

(ii) The mathematical model represents the qualitative characteristics of the system (e.g., Fig. 2), but it is unclear to what extent the model fits / predictions are quantitatively consistent with the experimental data. An expansion of the analysis by suitable goodness-of-fit tests (given the uncertainty in the data) is suggested. In addition, a direct comparison of model predictions and experimental data for the two-input cases (separated in Fig. 2f,g) would facilitate an assessment of model quality. These aspects are important to support detailed statements on the model such as 'Hill kinetics suit modeling of DD domain responses'.

(iii) The *in silico* feedback experiments for set-point control appear limited in the current version in several aspects: (i) The fine tunability with control inputs (one main point in the characterization of the system) is not exploited, and hence it is not shown that / how this feature contributes to control performance. (ii) A single control target (50% protein abundance) appears insufficient to demonstrate general superiority of the approach, compared to previous studies. (iii) The claim of 'generating more complex time varying gene expression dynamics' therefore has limited support. (iv) Probably most importantly, the setup of the experiments (e.g., Fig. 3) is such that conditions without control action (Fig. 3a,b) are compared with those in which the controller can effectively act (Fig. 3c) given the initial conditions and control target. Given that a main argument for dual-input control is the slow dynamics of purely expression-based control, and that the system's time constants are predominantly set by protein degradation, an experiment with control by a Dox as single input, but destabilized protein (i.e., media without TMP in the setup in Fig. 3a) needs to be included in the comparison.

(iv) The control application for signaling in embryonic stem cells selects beta-catenin as an important component, but neither the construct used is new, nor are signaling effects of protein abundance control shown. Therefore, the claim of 'confirming full functionality of the conditional

protein' appears an overstatement – this part of the study would be significantly strengthened if it was shown that the external control achieves fine-tuned signaling behavior or physiological effects.

Minor comments:

(i) p.6, bottom: 'suitability for dynamic modulation' may be misleading for steady-state characterization.

(ii) p.9, 'Interestingly, when DD is present on both the transactivator (rtTA) and the fluorescent reporter, the residual protein stability ...' – would this behavior not be expected, and if not, how can it be explained?

(iii) p.10, 'possibly due to limited TMP efficacy in stabilizing both DDrtTA and DDmCherry.' – the basis for this statement is unclear, would this be due to limited intracellular TMP, competition for proteasomal capacity, or other factors? This clarification is especially important in view of the Discussion statements 'However, rates of AID-mediated protein degradation/recovery strongly depend on auxin uptake and metabolism, as well as on the abundance of SCF complex components, which might vary among different biological systems.'

(iv) Fig. 4a and related: It would facilitate data interpretation to show pdfs (instead of current normalization by maximum).

(v) Discussion, p.15: 'Our system suits generation of both spatial- and temporal- patterns of gene expression that traditionally have not been easy to manipulate.' – spatial control has not been shown and given that the control inputs are diffusible chemicals, it does not seem plausible.

(vi) Discussion, p.16: 'switch-like behaviour of our system' – please check against claims of fine tunability.

(vii) SI, p.45: 'The parameters were separately identified for EF1a-rtTA (mCherry and DDmCherry) and EF1a-DDrtTA (mCherry and DDmCherry) mESCs; only the Doxy Michaelis- Menten constant (DB) was directly fixed from dose-response experimental data.' – why were parameters of shared components (e.g., protein stabilities) not fixed to be identical for all constructs, and to what extent is the model then generalizable to new constructs or systems?

(viii) SI, p.45: Notation: γ is used both for the parameter vector, and for a subset of individual parameters.

Reviewer #2 (Remarks to the Author):

In this manuscript, Pedone and colleagues described the design and characterization of a dual system for controlling gene expression. The system combined a third generation Tetracycline-Inducible System (Tet-ON 3G) for transcriptional regulation and a Destabilising Domain (DD) from ecDHFR for targeted protein degradation.

Using microfluidic-based in-silico feedback control and other assays, the authors showed that the new system allows greater control of both protein dynamics and expression dynamic range over the individual technologies alone.

The data are of high quality and documented clearly. I think that this study will appeal to a wide variety of researchers who are engaged in molecular cell biology.

Points:

(1) In this paper, the authors used mCherry as a reporter and generated stable cell lines using mouse embryonic stem cells. It would be good to see that the system not only work in one cell line but work equally well in another cell type.

(2) The experiments involving fusing the rtTA with a Destabilising Domain do not seem to add much to the manuscript. This EF1a-DDrtTA was not even examined in a different protein such as beta-catenin.

(3) In addition to mCherry, the authors used mCherry-beta-catenin (S33Y) to examine the response of the dual-input system. As practical use in many studies will involve untagged proteins, it is also important to demonstrate the regulation of non-mCherry-tagged proteins. It is also important to demonstrate the system is able to tune the expression of a protein close to its physiological level. Another important demonstration of the usefulness of the system is to have a biological readout instead of just protein expression (e.g. what threshold of a protein is needed to elicit a certain functional response).

(4) Some indications of how heterogenous (or homogenous) is the response at individual cells level are needed.

Reviewers' comments:

Reviewer #1 (Remarks to the Author):

The manuscript by Pedone et al. Entitled ,A tunable dual-input system for 'on-demand' dynamic gene expression regulation' describes a synthetic biology system relying on independent control inputs for gene expression and protein stability and its application to dynamic control of protein abundances in mammalian cells. Compare to control of gene expression alone, the authors argue that their system decreases response times, enhances tunability and dynamic range, and improves controllability in in silico feedback applications. Proof-of-principle application include the control of signaling components in stem cell with potential future biomedical applications.

While many synthetic biology systems for gene expression control in mammalian systems have been developed, relatively few systems for the control of protein stability (e.g., via small molecules such as rapamycin) exist, and the combination of both approaches with orthogonal controllability is novel. A strength of the manuscript is the careful experimental characterization of the system combined with mathematical modeling. The proof-of-principle applications of in silico feedback also demonstrate feasibility in the complex stem cell setting.

In several aspects, however, improvements of the manuscript appear warranted, primarily in clarifying the advances over the state of the art, support for some claims, and application perspectives. More specifically:

(i) The basic design of the system relies on previously established and characterized components (tTA and DD-fusions), such that it is unclear to what extent the basic experimental characterization (Figs. 1,2) reveals novel (unexpected) features. A more detailed discussion/comparison with features of the original (related) control systems should be provided, for example, regarding robustness, specificity, precision, mechanisms of proteasomal degradation, steady-state characteristics.

We thank the Reviewer for this comment. It is absolutely true that our design relies on systems already proposed in the literature for transcriptional or post-translational only gene expression regulation (strengths and limitations of them had already been discussed in the Introduction, lines 47-74 of the revised Main text); however their combination had not been fully explored previously. We have added a sentence in the Discussion to specify this point (see revised Main Text, lines 425-427).

In our work, we believe to demonstrate, with a combination of modelling and experiments (some already present in the previous version of the manuscript, and

some new to address the Reviewer's comments, see specific replies below), that our dual-input system is superior in terms of:

1) Steady-state characteristics and dynamic range: the exclusive control of transcription might limit the achievable dynamic range when modulating a very stable protein. This is the case of β -catenin, that we previously showed to have >7 hours half life (Marucci et al., Cell Reports 2014), and that we managed to regulate also at protein stability level, both when DDmCherry-fused (Fig. 4a-e, Supplementary Fig. 4c-e, g, j-m), and when DD-fused only (new experiments, Fig. 4f, h; Supplementary Fig. 4r, s).

On the other hand, protein stability control alone also has some limitations: the strength of the promoter might hamper the effectiveness of the destabilising domain. Indeed, we showed that degron-tagged proteins still retain a certain stability (Supplementary Figs. 2b, q, x, 4g, l; Doxy1000ng/mL no TMP samples). Based on similar considerations, a modular library of degrons has been very recently proposed in the same journal to modulate half-life of mammalian proteins in a predictable manner (Chassin et al. Nat Commun. 2019). As compared to this work, our system, thanks to conditional protein regulation, allows modulation of protein stability without requiring the engineering of different constructs and excessive manipulation of the cell lines carrying them. We have now added a comment in the Discussion to compare our results with those in Chassin et al. (see Discussion, lines 441-446 of the revised Main text).

In our work, model simulations and experimental validation confirmed the ability of tightly controlling the exogenous protein steady-state characteristics and dynamic range when modulating both transcription and protein stability.

In the previous version of the manuscript, this had been proved in murine cells using a fluorescent DD-fused protein (Fig. 2 b, c, f, g; Supplementary Figs. 2 b, and 4g); we have now extended these results to DD-fused proteins not tagged with fluorescent reporters (Fig. 4h, i), and also for gene expression modulation in different cell backgrounds (Supplementary Fig. 4l) and/or types (Supplementary Fig. 2x).

2) Kinetic response in switch on/off dynamics: we already showed both in flow-cytometry time courses (compare Fig. 2d with e; Supplementary Fig. 2c with d), and in a microfluidics/microscopy setting (compare Fig. 3a; movie 1 with Fig. 3c; movie 3), that the switch-off kinetic of a traditional Tet-On system are much slower. Similar considerations hold when perturbing only protein degradation (compare Fig. 3a; movie 1 with Fig. 3b; movie 2). Instead, the combination of transcriptional and post-translation control minimises these issues enabling faster dynamic response of different proteins, fluorescent tagged or not and across different cell line (Fig. 4j, k; Supplementary Fig. 2y, z), also when cells are cultured and controlled in a microfluidics/microscopy platform (Figs. 3c, e, 4e).

Additionally, in a set of new experiments, we measured cell-to-cell variability in single cell induced response, and found that, as compared to a Tet-on system, it is reduced

when combining transcriptional and post-translational control (Supplementary Fig. 2e-g). These new results, suggesting more robust gene expression induction, are now discussed in the Main text (see lines 187-195 and lines 447-456 of the revised Main text).

All in all, we believe that we have now better proved and discussed the usefulness of our dual-input system for systems and synthetic biology applications across various genes of interests, cell lines and culture platforms.

(ii) The mathematical model represents the qualitative characteristics of the system (e.g., Fig. 2), but it is unclear to what extent the model fits/predictions are quantitatively consistent with the experimental data. An expansion of the analysis by suitable goodness-of-fit tests (given the uncertainty in the data) is suggested.

We agree with the Reviewer that the accuracy of the model had not been properly characterised. We have now calculated the Root Mean Squared Error (RMSE) to quantify the model consistency with data, (see legends of Figs 2a-e; Supplementary Fig. 2j-o and Supplementary Information). The RMSE order of magnitude for fitting results are similar among systems, and the same holds for model validations. Of note, the mathematical models have been simplified (3 ODEs each) to reduce the parameter space, and the fitting has been refined to improve consistency across the corresponding engineered systems, see response to the specific Reviewer point below. We thus believe that the models we developed, although not including all aspects of the biological systems (e.g. stochastic noise, diffusion and decay time of inducers etc.), can recapitulate with acceptable precision the key dynamic features, and discriminate well differences between single and double-input gene expression regulation observed experimentally.

In addition, a direct comparison of model predictions and experimental data for the two-input cases (separated in Fig. 2f,g) would facilitate an assessment of model quality. These aspects are important to support detailed statements on the model such as ‘Hill kinetics suit modeling of DD domain responses’.

To facilitate the comparison between model predictions and data, we now plot in Fig. 2g and Supplementary Fig. 2u, w the experimental validations next to the correspondent model predictions shown in Fig. 2f and Supplementary Fig. 2t, v; we hope it is easier to appreciate the good predictive power of the model.

(iii) The in silico feedback experiments for set-point control appear limited in the current version in several aspects: (i) The fine tunability with control inputs (one main point in the characterization of the system) is not exploited, and hence it is not shown that/how this feature contributes to control performance. (ii) A single control target (50% protein abundance) appears insufficient to demonstrate general superiority of the approach, compared to previous studies. (iii) The claim of ‘generating more complex time varying gene expression dynamics’ therefore has limited support.

Firstly, to quantitatively measure the control performance, we have now computed various control performance indexes for all control experiments (Integral of Squared Error, Integral Absolute Error, Integral Time-weighted Absolute Error, see Table 3, Supplementary Information).

To complete this set of experiments, we performed microfluidics/microscopy experiments with a time-varying, multi set-point reference, using EF1a-rtTA_TRE3G-DDmCherry mESCs and both Doxy and TMP as control input (see Fig. 3e; movie 5), and showed a good ability to regulate DDmCherry levels with performance indexes comparable to the 50% set-point reference (Fig. 3c; movie 3). See description of these results in the Main text (see lines 305-322 of the revised Main text)

Of note, in all these experiments, oscillations around the set-points are present; this is inevitable using a Relay control strategy and a relatively long sampling time (60 minutes to avoid phototoxicity, Supplementary Information), and had been previously reported using a comparable control platform (Fracassi et al. ACS Synth Biol 2016). More advanced control strategies, such as Model Predictive Control that we recently employed (Postiglione et al. ACS Synth Biol 2018) can improve the control performance; however, such controllers require a mathematical model of the system, not always available (this has now been discussed in the Main text, lines 500-504). We believe that engineering an inducible system more easily controllable, even with simple control strategies, can be advantageous; being the focus of this work the comparison of differently inducible systems rather than an evaluation and optimisation of different control strategies, a simple Relay approach only has been employed.

(iv) Probably most importantly, the setup of the experiments (e.g., Fig. 3) is such that conditions without control action (Fig. 3a,b) are compared with those in which the controller can effectively act (Fig. 3c) given the initial conditions and control target. Given that a main argument for dual-input control is the slow dynamics of purely expression-based control, and that the system's time constants are predominantly set by protein degradation, an experiment with control by a Dox as single input, but destabilized protein (i.e., media without TMP in the setup in Fig. 3a) needs to be included in the comparison.

We thank the Reviewer for suggesting this important experiment. We performed a 50% set-point Relay experiment using Doxy as control input on cells that have never been treated with TMP (Fig. 3d; movie 4), and we found a comparable control performance as when controlling the system with both inducers (see Fig. 3c, d; movies 3, 4 and Table 3, Supplementary Information). This result indicates that controlling only at transcriptional level a destabilised protein is a good strategy for Relay set-point regulation. Nevertheless, when testing the control performance with a time-varying, multi set-point staircase control reference, the control performance of the system was superior when applying both transcriptional and post-translational control (see Fig. 3e, f; movies 5, 6 and Table 3, Supplementary Information). In absence of TMP, the maximal EF1a-rtTA_TRE3G-DDmCherry mESC induction with Doxy is lower (Supplementary Fig. 2d); indeed, in microscopy/microfluidics

experiments, we had to change the microscope settings to appreciate a good fluorescence signal, comparable to Doxy/TMP induced cells and informative for online protein quantification (Supplementary Information). Therefore, while the fluorescence signal is normalised to 1 as maximal induction fluorescence (calculated during the calibration phase, 120min), the actual absolute values are lower for the control experiments in Fig. 3d, f as compared to those in Fig. 3 c, e; this might explain the different slope in the dynamic response to Doxy addition/removal, with consequent more evident deviation from the reference and worst control performance in Fig. 3f.

We present and discuss these new results in the Main text (see lines 299-322 of the revised Main text).

(iv) The control application for signaling in embryonic stem cells selects beta-catenin as an important component, but neither the construct used is new, nor are signaling effects of protein abundance control shown. Therefore, the claim of ‘confirming full functionality of the conditional protein’ appears an overstatement – this part of the study would be significantly strengthened if it was shown that the external control achieves fine-tuned signaling behavior or physiological effects.

To study the functionality and physiological effect of inducible exogenous β -catenin protein, we took advantage of a mESC line deleted for endogenous β -catenin gene we generated (β -catenin^{-/-} mESCs; Aulicino et al. bioRxiv 2019; doi.org/10.1101/661777), and tested if and how we could rescue the reported impaired clonogenicity and survival of β -catenin null mESCs grown in absence of the Leukemia Inhibitory Factor -LIF- (Wray et al. Nat Cell Biol 2011; Lyashenko et al. Nat Cell Biol 2011).

We stably transduced β -catenin^{-/-} mESCs with the EF1a-rtTA_TRE3G-DDmCherry β -catenin^{S33Y} construct (henceforth C1-EF1a-rtTA_TRE3G-DDmCherry β -catenin^{S33Y} mESCs). Firstly, we checked functionality of the inducible construct. As with previous constructs, we found selective dose-response of exogenous DDmCherry β -catenin^{S33Y} mRNA (Supplementary Fig. 4h, i) and protein (Supplementary Fig. 4j-l) levels to Doxy and Doxy/TMP treatment, respectively. We also measured the levels of total, cytosolic and nuclear β -catenin^{S33Y} protein upon induction by western-blot (Supplementary Fig. 4m). All extracts showed a dose-dependent increase in the amount of the exogenous DDmCherry β -catenin^{S33Y} following Doxy/TMP treatment (Supplementary Fig. 4m), consistent with flow cytometry data (Supplementary Fig. 4l). As compared to the endogenous β -catenin amount, measured in wild-type mESCs, only in the nuclear extracts DDmCherry β -catenin^{S33Y} levels are above the endogenous ones (Supplementary Fig. 4m, Nuclear); this observation is coherent with the stabilised protein form we used.

Then, to investigate the effect of DDmCherry β -catenin^{S33Y} over-expression, we performed a clonogenicity assay in 5 different culture conditions following LIF withdrawal. C1-EF1a-rtTA_TRE3G-DDmCherry β -catenin^{S33Y} mESCs were exposed

to TMP10 μ M and increasing concentrations of Doxy (10-100ng/mL; Supplementary Fig. 4n), and pluripotent colonies were detected by Alkaline Phosphatase (AP), which only stains cells retaining the pluripotent phenotype. When uninduced cells were cultured in serum- and LIF-free media, the number of AP positive cells was null or the signal was faint, while in NDiff-Chiron/PD, NDiff-Chiron and NDiff-PD media supplemented with TMP10 μ M and Doxy10ng/mL, but not Doxy100ng/mL, some AP+ colonies appeared, rescuing the impaired clonogenicity (Supplementary Fig. 4o).

These results, while confirming the previous observation of LIF dependency in β -catenin-deficient mESCs, indicate that levels of exogenous β -catenin need to be kept within a certain threshold for rescuing pluripotency, and further justify the need of tools to precisely modulate gene expression in mammalian cells.

These new results are discussed in the main text (see lines 365-398 and lines 506-509 of the revised Main text).

Minor comments:

(i) p.6, bottom: ‘suitability for dynamic modulation’ may be misleading for steady-state characterization.

In that section, we also reported switch-on and -off data. We tried to make this point clearer within the text, please see lines 143-147 of the revised Main text.

(ii) p.9, ‘Interestingly, when DD is present on both the transactivator (rtTA) and the fluorescent reporter, the residual protein stability ...’ – would this behavior not be expected, and if not, how can it be explained?

The Reviewer is right, this result is not surprising as, in absence of TMP, both the transactivator and the fluorescent reporter are destabilised resulting in lower DDmCherry expression, we have rephrased this sentence, see lines 214-217 of the revised Main text.

(iii) p.10, ‘possibly due to limited TMP efficacy in stabilizing both DDrtTA and DDmCherry.’ – the basis for this statement is unclear, would this be due to limited intracellular TMP, competition for proteasomal capacity, or other factors? This clarification is especially important in view of the Discussion statements ‘However, rates of AID-mediated protein degradation/recovery strongly depend on auxin uptake and metabolism, as well as on the abundance of SCF complex components, which might vary among different biological systems.’

We agree that the uptake of inducers can have an impact on all inducible systems; therefore, we have slightly rephrased the sentence related to AID-mediated protein degradation in the Discussion, see lines 462-464 of the revised Main text.

(iv) Fig. 4a and related: It would facilitate data interpretation to show pdfs (instead of current normalization by maximum).

We amended Figs. 1b, d, and 4a; Supplementary Figs. 2h, I and 5b as suggested, by showing the pdfs (number of cells) instead of the % of Max as in the previous version of the manuscript.

(v) Discussion, p.15: ‘Our system suits generation of both spatial- and temporal- patterns of gene expression that traditionally have not been easy to manipulate.’- spatial control has not been shown and given that the control inputs are diffusible chemicals, it does not seem plausible.

We agree that spatial control might require additional control components (e.g. optogenetics), and have removed it from the mentioned sentence, see lines 505-506 of the revised Main text.

(vi) Discussion, p.16: ‘switch-like behaviour of our system’ – please check against claims of fine tunability.

We agree this claim was confusing, the have removed “switch-like” from that sentence, see line 519 of the revised Main text.

(vii) SI, p.45: ‘The parameters were separately identified for EF1a-rtTA (mCherry and DDmCherry) and EF1a-DDrtTA (mCherry and DDmCherry) mESCs; only the Doxy Michaelis- Menten constant (DB) was directly fixed from dose-response experimental data.’ – why were parameters of shared components (e.g., protein stabilities) not fixed to be identical for all constructs, and to what extent is the model then generalizable to new constructs or systems?

The Reviewer comment is correct; the modelling had been originally performed in parallel to experiments, done first with the EF1a-rtTA and EF1a-DDrtTA systems and, later, using the EF1a-DDrtTA_TRE3G-mCherry and EF1a-DDrtTA_TRE3G-DDmCherry constructs.

In the revised version of the manuscript, we have refined the models and parameters to keep, as much as possible, parameters of shared components fixed/comparables across constructs.

Firstly, to reduce the parameter space, we simplified the 4 models (before composed of 4 ODEs each) by considering the rtTA mRNA concentration at steady state; of note, this variable (for which an ODE was previously included) does not depend on the system inputs (i.e. Doxy and TMP) or on other variables, so assuming it is a steady-state allows decreasing by 2 the size of the parameter space, while not altering the overall model structure.

The new 4 systems of ODEs for the EF1a-rtTA_TRE3G-mCherry, EF1a-rtTA_TRE3G-DDmCherry, EF1a-DDrtTA_TRE3G-mCherry and EF1a-DDrtTA_TRE3G-DDmCherry mESCs were then fitted to data using, as previously, the Matlab function `fmincon` to minimise the cost objective function (Supplementary Information) and using the following approach:

i) for all constructs, the Doxy and TMP Michaelis-Menten constants were directly fixed from dose-response experimental data (Fig. 2a-c; Supplementary Fig. 2a, b; Supplementary Fig. 2j-m, p, q);

ii) for all constructs, the degradation of mCherry protein (untagged, or in presence of saturating TMP for the DD-tagged forms) was fitted to be comparable to that measured experimentally (Fig. 1f, g; Supplementary Fig. 1a, b);

iii) for all constructs, during the fitting procedure, a constraint on TMP-dependent degradation of the transactivator and/or DDmCherry protein was inserted so that, in presence of saturating TMP concentration, it would be equal to the degradation rate of the untagged mCherry proteins;

iv) for the EF1a-DDrtTA_TRE3G-mCherry and EF1a-DDrtTA_TRE3G-DDmCherry systems (equations (7)-(9), and (10)-(12), respectively, Supplementary Information), we fixed a number of parameters as in the EF1a-rtTA systems, specifically: the Hill coefficients for Doxy (h_1) and for TRE3G promoter (h_2), the TRE3G Michaelis-Menten constant (K_2) and degradation rate of mCherry mRNA (d_3).

Also, for EF1a-DDrtTA_TRE3G-DDmCherry mESCs, parameters related to TMP-dependent degradation were kept identical in the transactivator and DDmCherry protein equations. In such a way, the vast majority of parameters were kept identical, or of the same order of magnitude, across the 4 systems.

The parameters that varied the most between the EF1a-rtTA_TRE3G-mCherry/TRE3G-DDmCherry, and the EF1a-DDrtTA_TRE3G-mCherry/TRE3G-DDmCherry systems are those that might depend on plasmid copy number (i.e. production of rtTA protein, degradation rate of rtTA protein, basal activity of TRE3G, maximal transcription rate of TRE3G, production rate of mCherry protein); we would expect them to change, given that the EF1a-rtTA_TRE3G-mCherry/TRE3G-DDmCherry, and the EF1a-DDrtTA_TRE3G-mCherry/TRE3G-DDmCherry lines come from 2 different groups of cells (i.e. firstly stably transduced with EF1a-rtTA or EF1a-DDrtTA constructs, see “Cell Line Derivation” section in Material and Methods, Main text).

We therefore believe that the models we developed can be generalised at least in terms of structure; inevitably, a few parameters might need to be identified again if intended to describe dynamics of these (or related) constructs in different cell lines, to account for plasmid copy-number, transfection/infection efficacies and species-dependent transcription, translation and degradation machinery variations.

(viii) SI, p45: Notation: γ is used both for the parameter vector, and for a subset of individual parameters.

We have changed the parameter vector name to p in Supplementary Information.

Reviewer #2 (Remarks to the Author):

In this manuscript, Pedone and colleagues described the design and characterization of a dual system for controlling gene expression. The system combined a third generation Tetracycline-Inducible System (Tet-ON 3G) for transcriptional regulation and a Destabilising Domain (DD) from ecDHFR for targeted protein degradation.

Using microfluidic-based in-silico feedback control and other assays, the authors showed that the new system allows greater control of both protein dynamics and expression dynamic range over the individual technologies alone.

The data are of high quality and documented clearly. I think that this study will appeal to a wide variety of researchers who are engaged in molecular cell biology.

Points:

(1) In this paper, the authors used mCherry as a reporter and generated stable cell lines using mouse embryonic stem cells. It would be good to see that the system not only work in one cell line but work equally well in another cell type.

Following the Reviewer's suggestion, we engineered HeLa cells with both EF1a-rtTA_TRE3G-mCherry and EF1a-rtTA_TRE3G-DDmCherry encoding plasmids. We measured a good dose-response of the reporter to varying levels of inducers (Supplementary Fig. 2x), and also faster switch-off dynamics upon inducer removal in EF1a-rtTA_TRE3G-DDmCherry as compared to EF1a-rtTA_TRE3G-mCherry HeLa cells (Supplementary Fig. 2y, z). We therefore believe that our dual-input system can be generalised for tuning gene expression across mammalian cell lines. Please see discussion of new results in the Main text, lines 248-268.

(2) The experiments involving fusing the rtTA with a Destabilising Domain do not seem to add much to the manuscript. This EF1a-DDrtTA was not even examined in a different protein such as beta-catenin.

As in the main text, we had hypothesised that the addition of a degron on both the transactivator and the controlled gene could further reduce the time to switch-off the system. There is to say that both EF1a-DDrtTA_TRE3G-mCherry and EF1a-DDrtTA_TRE3G-DDmCherry mESCs switched-off the fluorescent reporter upon inducer removal faster than cells carrying the standard Tet-on system (compare Supplementary Fig. 2n, o -dots representing data in Supplementary Fig. 2r, s- and Fig. 2d -dots representing data in Supplementary Fig. 2c). This, not properly discussed in the previous version of the manuscript, has now been specified (see lines 224-228 of the revised Main text).

It is also true that the DD presence on both the destabilising domain and mCherry did not further shorten protein switch-off kinetics, as compared to the EF1a-rtTA_TRE3G-DDmCherry system (compare data in Supplementary Fig. 2o and in Fig. 2e).

Because steady-state activation and switch-off were not improved in EF1a-DDrtTA_TRE3G-mCherry and EF1a-DDrtTA_TRE3G-DDmCherry mESCs we did not apply these systems to regulate β -catenin, and we performed all subsequent experiments with the EF1a-rtTA_TRE3G-DDmCherry construct.

Nevertheless, we think this comparison is important, also in light of recent results in (Chassin et al. Nat Commun 2019), where a modular library of degrons has been engineered to modulate half-life of mammalian proteins. Specifically, those degrons had been fused to a transactivator of Tet-ON/OFF systems; in agreement with our results, the authors showed that changing the transactivator half-life can modulate the dynamic range of the regulated gene. As compared to Chassin et al. work, our system, thanks to conditional protein regulation, allows modulation of protein stability without requiring the engineering of different constructs and cell lines carrying them. We have now added a comment in the Discussion to compare our results with those in Chassin et al. (see Discussion, lines 441-446 of the revised Main text)

(3) In addition to mCherry, the authors used mCherry-beta-catenin (S33Y) to examine the response of the dual-input system. As practical use in many studies will involve untagged proteins, it is also important to demonstrate the regulation of non-mCherry-tagged proteins.

We agree with the Reviewer. To show that our system can be used to modulate untagged proteins, we engineered two new constructs (TRE3G-DD β -catenin^{S33Y} and TRE3G-DDLef1) and stably transduced them in β -catenin^{-/-} or wild-type mESCs carrying the EF1a-rtTA construct (Fig. 4f, g, respectively). We found adequate and selective dose-response of mRNA (Supplementary Fig. 4p, q) and protein (Fig. 4h, i; Supplementary Fig. 4r-u) steady-state activation. Finally, we also measured the switch-off dynamic following inducers removal and found protein varied according to treatments (Fig. 4j, k) as reported for previous constructs.

See discussion of these new results in the main text, lines 399-412.

It is also important to demonstrate the system is able to tune the expression of a protein close to its physiological level.

To address this point, as well as the next one (see response below), we took advantage of a mESC line deleted for endogenous β -catenin gene we generated (β -catenin^{-/-} mESCs, Aulicino et al. bioRxiv 2019 doi.org/10.1101/661777), and stably transduced them with the EF1a-rtTA_TRE3G-DDmCherry β -catenin^{S33Y} construct (named C1-EF1a-rtTA_TRE3G-DDmCherry β -catenin^{S33Y} mESCs). We checked functionality of the inducible construct; as with previous constructs, we found selective dose-response of exogenous DDmCherry β -catenin^{S33Y} mRNA (Supplementary Fig. 4h, i) and protein (Supplementary Fig. 4j-l) levels to Doxy and Doxy/TMP treatment, respectively. We also measured the levels of total, cytosolic and nuclear β -catenin^{S33Y} protein upon induction by western-blot (Supplementary Fig. 4m). All extracts showed a dose-dependent increase in the amount of the exogenous DDmCherry β -catenin^{S33Y} following Doxy/TMP treatment (Supplementary Fig. 4m), consistent with flow cytometry data (Supplementary Fig. 4l). As compared

to the endogenous β -catenin amount, measured in wild-type mESCs, only in the nuclear extracts DDmCherry β -catenin^{S33Y} levels are above the endogenous ones (Supplementary Fig. 4m, Nuclear); this observation is coherent with the stabilised protein form we used, which is insensitive to the endogenous degradation machinery and therefore accumulates into the nucleus more than the native β -catenin. These new results are discussed in the Main text, lines 365-383.

We would like to remark that different results might arise when modulating a gene with different physiological levels (for example, due to different strength of the endogenous promoter, or different native protein stability). The endogenous β -catenin is a very stable and highly expressed protein in mESCs (Marucci et al., Cell Reports 2014); therefore it is not surprising that, in a β -catenin null background, we do not reach the wild-type levels of total protein.

Still, the modularity of our system could allow reaching higher/lower maximal induction (and so, fold change as compared to physiological protein level) varying the copy number of the transduced plasmids carrying the transactivator and/or the gene of interest.

Another important demonstration of the usefulness of the system is to have a biological readout instead of just protein expression (e.g. what threshold of a protein is needed to elicit a certain functional response).

To study the functionality and threshold effect of inducible exogenous β -catenin protein, we performed experiments with the aforementioned C1-EF1a-rtTA_TRE3G-DDmCherry β -catenin^{S33Y} mESCs we generated. Specifically, we tested if and how inducible DDmCherry β -catenin^{S33Y} was able to rescue the reported impaired clonogenicity and survival of β -catenin null mESCs grown in absence of the Leukemia Inhibitory Factor -LIF- (Wray et al. Nat Cell Biol 2011; Lyashenko et al. Nat Cell Biol 2011).

We performed a clonogenicity assay in 5 different culture conditions following LIF withdrawal. C1-EF1a-rtTA_TRE3G-DDmCherry β -catenin^{S33Y} mESCs were exposed to TMP10 μ M and increasing concentrations of Doxy (10-100ng/mL; Supplementary Fig. 4n), and pluripotent colonies were detected by Alkaline Phosphatase (AP), which only stains cells retaining the pluripotent phenotype. When uninduced cells were cultured in serum- and LIF-free media, as previously reported (Wray et al. Nat Cell Biol 2011; Lyashenko et al. Nat Cell Biol 2011), the number of AP positive cells was null or the signal very faint, while in NDiff-Chiron/PD, NDiff-Chiron and NDiff-PD media supplemented with TMP10 μ M and Doxy10ng/mL, but not Doxy100ng/mL, some AP+ colonies appeared, rescuing the impaired clonogenicity (Supplementary Fig. 4o).

These results, while confirming the previous observation of LIF dependency in β -catenin-deficient mESCs, indicate that levels of exogenous β -catenin need to be kept within a certain threshold for rescuing pluripotency, as excessive amounts most likely

could instead promote cells differentiation, and further justify the need of tools to precisely modulate gene expression in mammalian cells.

These new results are discussed in the main text, lines 384-398.

(4) Some indications of how heterogenous (or homogenous) is the response at individual cells level are needed.

To assess the response to drugs in our dual-input system at single cell level, we analysed the switch-on and -off transitions of 13 individual clones FACS-sorted from both EF1a-rtTA_TRE3G-mCherry and EF1a-rtTA_TRE3G-DDmCherry mESC populations (Supplementary Fig. 2e, f, respectively). Both the switch-on and -off response of EF1a-rtTA_TRE3G-DDmCherry cells was more homogenous across clones when compared to EF1a-rtTA_TRE3G-mCherry cells, as indicated by measuring the coefficient of variation of mCherry expressing cells (Supplementary Fig. 2 g), suggesting that conditional control of protein stability might reduce cell-to-cell variability.

See description of the new results in the Main text, lines 187-195, and discussion lines 447-456.

REVIEWERS' COMMENTS:

Reviewer #1 (Remarks to the Author):

The revision addresses the comments from the first round of review very well. Especially the new experiments on multi-setpoint control and beta-catenin control substantially strengthen the manuscript's generality and applicability, respectively. The revised mathematical models achieve higher consistency, and likely make them applicable for future studies.

Minor comments (suggested changes in []):

(i) l.184: 'the ... protein ... has [lower] half-life [] mCherry'

(ii) l. 402: '...expressing the transactivator and [a] the conditionally...'

Reviewers' comments:

Reviewer #1 (Remarks to the Author):

The revision addresses the comments from the first round of review very well. Especially the new experiments on multi-setpoint control and beta-catenin control substantially strengthen the manuscript's generality and applicability, respectively. The revised mathematical models achieve higher consistency, and likely make them applicable for future studies.

Minor comments (suggested changes in []):

(i) l.184: 'the ... protein ... has [lower] half-life [] mCherry'

The sentence has been changed according to the Reviewer's suggestion, see line 178 of the main text.

(ii) l. 402: '...expressing the transactivator and [a] the conditionally...'

The sentence has been changed according to the Reviewer's suggestion, see line 393 of the main text.